



# Aura/MLS observes, and SD-WACCM-X simulates the seasonality, quasi-biennial oscillation and El Nino Southern Oscillation of the migrating diurnal tide driving upper mesospheric CO primarily through vertical advection

Cornelius Csar Jude H. Salinas[1,2], Dong L. Wu[3], Jae N. Lee[3,4], Loren C. Chang[1,2], Liying Qian[5] and Hanli Liu[5]

[1]Department of Space Science and Engineering, National Central University, Taoyuan City, 32001, Taiwan
[2]Center for Astronautical Physics and Engineering, National Central University, Taoyuan City, 32001, Taiwan
[3]NASA Goddard Space Flight Center, Greenbelt, Maryland, 20771, USA
[4]Joint Center for Earth Systems Technology, University of Maryland, Baltimore County, Baltimore, Maryland, 21201, USA
[5]NCAR High Altitude Observatory, Boulder, Colorado, 80301, USA

*Correspondence to*: Cornelius Csar Jude H. Salinas (ccjsalinas@gmail.com)

**Abstract.** This work uses 17 years of upper mesospheric carbon monoxide (CO) and temperature observations by the Microwave Limb Sounder (MLS) on-board the Aura satellite to present and explain the seasonal and interannual variability of

the migrating diurnal tide (DW1) component of upper mesospheric CO. This work then compares these observations to simulations by the Specified Dynamics – Whole Atmosphere Community Climate Model with Ionosphere/Thermosphere eXtension (SD-WACCM-X). Results show that, for all seasons, MLS CO DW1 peaks above 85 km and has a latitude structure resembling the (1,1) mode in temperature. On the other hand, SD-WACCM-X DW1 also peaks above 85 km and has a latitude structure resembling the (1,1) mode but it simulates 2 local maximum of the (1,1) mode between 85 km and 92 km. Despite

the differences in altitude structure, a tendency analysis and the adiabatic displacement method revealed that, on seasonal and interannual timescales, observed and modelled CO's (1,1) component can be reproduced solely using vertical advection. It was also found that both observed and modelled CO's (1,1) component contains interannual oscillations with periodicities close to that of the Quasi-biennial Oscillation and the El Nino Southern Oscillation. From these results, this work concludes that on seasonal and interannual timescales, the observed and modelled (1,1) mode affects the global structure of upper

mesospheric CO primarily through vertical advection.

## 1 Introduction

The most dominant chemical reaction driving upper mesospheric Carbon Monoxide (CO) is the photo-dissociation of carbon dioxide ($CO_2$) by solar ultraviolet (UV) radiation (Brasseur and Solomon, 2006):

$$CO_2 + h\nu \rightarrow CO + O \tag{1}$$





This reaction makes CO's photochemical lifetime longer than dynamical timescales (Minschwaer et al, 2010). Thus, numerous studies have used CO as a dynamical tracer in the upper mesosphere particularly for the winter residual circulation (Allen et al, 1999; 2000; Manney et al, 2009; Lee et al, 2011; Garcia et al, 2014).

While numerous studies have used CO as a dynamical tracer for the winter residual circulation, nobody has used CO as a dynamical tracer for atmospheric tides. The upper mesosphere is a region where atmospheric tides reach significant

amplitudes. The most dominant atmospheric tide is the migrating diurnal tide. For all latitudes and altitudes, the migrating diurnal tide manifests as a westward propagating planetary-scale wave with zonal wavenumber 1 and with a period of 24 hours. The common nomenclature for the migrating diurnal tide is DW1. D stands for diurnal or its 24-hour period, W stands for westward which is its propagation direction and the number 1 stands for its zonal-wavenumber. For semidiurnal tides, we replace D with S and for eastward propagating tides, we replace W with E. For example, the eastward propagating non-

migrating semidiurnal tide with wavenumber 2 is written as SE2. While the manifestation of DW1 at all latitudes and altitudes is the same in terms of longitudinal propagation direction, wavenumber, and period, they can differ in terms of tidal amplitude and phase. The altitude and latitude variation of a tide's amplitude and phase determines what global tidal mode is currently present in the atmosphere. Hough modes mathematically represents global tidal modes (Chapman and Lindzen, 1970; Forbes, 1995). Classical tidal theory derives the Hough modes (Chapman and Lindzen, 1970). The most dominant tidal mode behind

the migrating diurnal tide in the upper mesosphere is the (1,1) Hough mode, a symmetric vertically propagating mode. This mode is generated by tropospheric water vapor and stratospheric ozone's absorption of solar radiation as well as by latent heat release from tropical convection (Grove, 1982a; Grove, 1982b; Hagan and Forbes, 2002). It then propagates from the lower atmosphere up to the mesosphere and lower thermosphere (MLT) region where its amplitude peaks before it breaks and dissipates.

Numerous observational studies have already shown that the DW1 component of temperature exhibits significant seasonal and interannual variability. A semi-annual oscillation with primary peaks in March equinox dominates their seasonal variability (Zhang et al, 2006; Forbes and Wu, 2006; Mukhtarov et al, 2009; Gan et al, 2014). Observations have also shown that the Quasi-Biennial Oscillation and the El Nino Southern Oscillation affects DW1 amplitudes (Lieberman, 1997; Vincent et al., 1998; McLandress, 2002a; 2002b; Gurubaran et al, 2005; Mayr & Mengel, 2005; Liebermann et al, 2007; Wu et al,

2008; Gurubaran et al., 2009; Mukhtarov et al, 2009; Pancheva et al, 2009; Xu et al., 2009; Pedatella et al, 2012; Pedatella et al, 2013; Gan et al, 2014; Liu et al, 2017; Zhou et al, 2018; Kogure et al, 2021; Pramitha et al, 2021; Cen et al, 2022). On the other hand, minimal observational studies have analysed the seasonal and interannual variability of the DW1 component of other dynamical and chemical parameters because of the lack of long-term reliable observations. To the best of our knowledge, Wu et al (2008) is the only observational study to show that there is also a possible QBO variation in the DW1 component of

horizontal winds. With regards tracers, previous studies have only shown that DW1 can affect the volume mixing ratio of tracers through transport processes (Akmaev et al, 1980; Angelats I Coll and Forbes, 1998; Marsh et al, 1999; Shepherd et al, 1995; Shepherd et al, 1997; Ward, 1999; Zhang et al, 1998; Marsh and Russell, 2000; Oberheide and Forbes, 2008; Smith et al, 2010; Marsh, et al, 2011; Salinas et al, 2020; Salinas et al, 2022). However, no observational study has analysed the seasonal





and interannual variabilities in a tracer's DW1 component. Thus, our knowledge of DW1-induced tracer transport is terribly

lacking. This hinders us from fully understanding atmosphere-ionosphere coupling in seasonal and interannual timescales
which highly depends on the transport of tracers like atomic oxygen (Jones et al, 2014).

This work helps remedy this issue by taking advantage of 17 years of CO observations provided by the Microwave
Limb Sounder (MLS) on-board the Aura satellite to analyse the seasonal and interannual variability of the DW1 component
of upper mesospheric CO. This work then compares these observations to simulations by the Specified Dynamics – Whole

Atmosphere Community Climate Model with Ionosphere/Thermosphere eXtension (SD-WACCM-X).

## 2. Satellite Datasets and Model

This work uses Aura MLS version 4.2x (V4.2x) carbon monoxide (CO) volume-mixing ratio and temperature vertical
profile observations from 2004 to 2021 (https://mls.jpl.nasa.gov/data/v4-2_data_quality_document.pdf). The MLS CO profiles
have a vertical resolution of 3-4 km in the stratosphere and lower mesosphere as well as 9 km in the upper mesosphere. With

this vertical resolution, MLS CO vertical profiles have 37 data-points from the stratosphere to the upper mesosphere. The MLS
temperature profiles have a vertical resolution of 4-6 km in the stratosphere and lower mesosphere as well as 8–13 km in the
upper mesosphere. With this vertical resolution, MLS temperature vertical profiles have 55 data-points from the stratosphere
to the upper mesosphere. This work uses these temperature observations to explain CO DW1. More details on this later. The
ascending nodes of the Aura orbit, when the spacecraft is moving toward the north, cross the equator at 1:45±15 PM (short-

handed to 2PM hereafter) local time. Similarly, the descending nodes, when the spacecraft is moving toward the south, cross
the equator at 1:45±15 AM (short-handed to 2AM hereafter) local time (https://aura.gsfc.nasa.gov/scinst.html). This orbit
allows MLS to supply near-global maps of 2AM and 2PM CO and temperature from the stratosphere to the upper mesosphere.
This sampling hinders us from getting the exact amplitude and phase of any CO and temperature tidal component. However,
this sampling still allows us to get the DW1-induced perturbations on CO (hereafter referred to as MLS CO $\mu'$) and temperature

(hereafter referred to as MLS $T'$). To calculate the DW1-induced perturbations, we take the difference of the 2AM and 2PM
zonal-mean profiles (Oberheide et al, 2003).

These observations are compared to simulations from SD-WACCM-X. WACCM-X is a first-principles Physics-
based model that simulates the whole atmosphere from the surface to the Ionosphere/Thermosphere up to around 700 km
depending on solar activity while accounting for the coupling of the atmosphere with the ocean, sea ice and land. It uses

elements of both the Whole Atmosphere Community Climate Model and the Thermosphere Ionosphere Electrodynamics
general circulation models (Liu, H.L. et al ,2018; Liu, J., et al, 2018). This work uses the Specified Dynamics mode of the
model (SD-WACCM-X). SD-WACCM-X is a version of WACCM-X whose temperature and winds from the surface to the
stratosphere at ~50 km is nudged by the Modern-Era Retrospective Analysis for Research and Applications (MERRA)
Reanalysis dataset (Rienecker et al, 2011; Marsh et al, 2013). By nudging with MERRA, the model's dynamical variables

become realistic from the surface to the stratosphere (Kunz et al, 2011). The run has a conventional latitude-longitude grid
with horizontal resolution of 1.9° in latitude and 2.5° in longitude. The vertical resolution is 2 points per scale height below





~50 km and increases to 4 points per scale height above ~50 km. Model parameters are output daily in hourly resolution. This work uses model outputs from 2004 to 2019. From these outputs, we calculate the monthly-means of dynamical and chemical parameters' the longitude-latitude-pressure-UT profiles. Then, from these 4-dimensional profiles, 2D least-squares fit is used

to calculate their zonal-mean component as well as the DW1 amplitudes and phases (Wu et al, 1995). From these DW1 amplitudes and phases, we reconstruct the zonal-mean profiles of all the parameters at 2AM and 2PM local-times. Finally, the DW1-induced perturbations are calculated by taking the difference of these 2AM and 2PM zonal-mean profiles.

To analyse SD-WACCM-X CO's DW1 component, we first need to assess the model's simulations of CO's daily-mean zonal-mean component. This will specifically be important in assessing any differences and/or similarities between MLS

and SD-WACCM-X CO $\mu'$. An estimate of the daily-mean zonal-mean profile of MLS CO (hereafter referred to as MLS CO $\bar{\mu}$) is calculated by taking the average of the 2AM and 2PM zonal-mean profiles. As already mentioned above, the daily-mean zonal-mean profile of SD-WACCM-X CO (hereafter referred to as SD-WACCM-X CO $\bar{\mu}$) is calculated using a 2D-least squares fit.

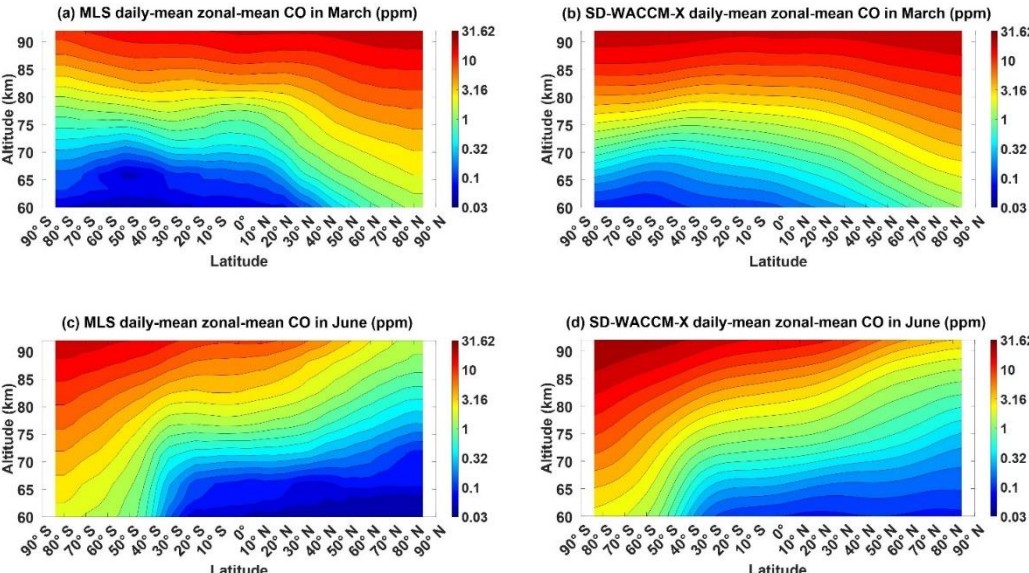

**Figure 1: Daily-mean zonal-mean component of (a) MLS CO in March equinox, (b) SD-WACCM-X CO in March equinox, (c) MLS CO in June solstice and (d) SD-WACCM-X CO in June solstice. All are in units of ppm.**

Figure 1 shows the CO $\bar{\mu}$ averaged for all March equinox and for all June solstice as observed by MLS and as simulated by SD-WACCM-X. In both seasons and in both observations and model, CO $\bar{\mu}$ shows interhemispheric asymmetry

with larger asymmetry during solstice seasons than during equinox seasons (September equinox and December solstice shown in figure A1). The interhemispheric asymmetry is characterized by larger CO over the winter (during solstice months) and/or spring (during equinox months) hemispheres than the summer and/or fall hemispheres, respectively. The latitudinal gradient over the winter hemisphere maximizes during solstice seasons. These seasonal morphologies are known to be driven by the seasonality of the residual circulation (Garcia et al, 2014). While these seasonal morphologies are similar in both observations


and model, there are still notable differences between them. During equinox seasons, MLS CO $\bar{\mu}$ above 80 km is larger over the equator than over the northern and southern low latitudes. Below 80 km, MLS CO $\bar{\mu}$ is lower over the equator than over the northern and southern low latitudes. This isn't reproduced in the model. This may be attributed to the incomplete local-time sampling in MLS. With incomplete local-time sampling, the zonal-mean may have aliases from other tides, particularly the semidiurnal tides. During solstice seasons, the latitudinal (vertical gradient) gradient over the winter hemisphere is larger

(weaker) in MLS CO than in SD-WACCM-X CO. This may be attributed to a weaker winter downwelling in the model than observed. These similarities and differences in MLS and SD-WACCM-X CO $\bar{\mu}$ will be noted in the analyses that follow.

To determine potential issues related to the lack of full local-time sampling in MLS temperatures, this work utilizes temperature observations from the Sounding of the Atmosphere using Broadband Emission Radiometry (SABER) instrument onboard the Thermosphere Ionosphere Mesosphere Energetics and Dynamics satellite (Russell et al, 1999). This work

specifically utilizes SABER v2.07 operational temperature profiles (Mertens et al, 2001; Kutepov et al, 2006; Garcia-Comas et al, 2008; Mertens et al, 2009; Dawkins et al, 2018). Salinas et al (2020) has pointed out that the assumption of no local-time variation in $CO_2$ may be problematic above 90 km, however, this work focuses on altitudes below 92 km where the $CO_2$ vertical gradient and thus local-time variations are negligible (Garcia et al, 2014). SABER has alternating latitudinal coverage of 82°N – 53°S and 53°N – 82°S that occur due to the spacecraft yaw cycle every ~60 days. The mission has an orbital period

of ~1.6 hours and a local time precession of 12 minutes per day. This orbit allows SABER to achieve full diurnal coverage after 60 days (Zhang et al, 2006). From these SABER temperature observations, we take all the profiles within 30-days before and after the 15th of every month, then bin them into a 30-degree longitude, 5-degree latitude, 2-km altitude and 3-hour UT grid. We then use 2D least-squares fit to calculate SABER temperatures' zonal-mean and DW1 component from these grids. From these DW1 amplitudes and phases, we reconstruct 2AM and 2PM zonal-mean temperature profiles. Finally, we apply

the same process as done on the MLS 2AM and 2PM zonal-mean temperature profiles to calculate the SABER DW1 temperature components. However, we only show values between latitudes 50°S and 50°N because SABER's yaw cycle hinders full local-time coverage at higher latitudes.

## 3. Seasonality of the DW1 Component of CO

In this section, we present the seasonality of the DW1 component of CO and temperature as observed by MLS and

as simulated by SD-WACCM-X. To calculate the seasonality of MLS CO $\mu'$ and $T'$, we first construct monthly mean 2AM and 2PM global zonal-mean profiles of CO and temperature. Each profile has a 4-degree latitude bin from latitude 84° S to latitude 84° N. We then calculate the composite seasonal average of these 2AM and 2PM zonal-mean profiles. Finally, we take the difference between the 2AM and 2PM zonal-mean profiles for each month. To calculate the seasonality of SD-WACCM-X CO $\mu'$ and $T'$, we first calculate the monthly-means of their longitude-latitude-pressure-UT profiles. Then, from

these 4-dimensional profiles, 2D least-squares fit is used to calculate their zonal-mean component as well as the DW1 amplitudes and phases. From these DW1 amplitudes and phases, we reconstruct their zonal-mean profiles at 2AM and 2PM





local-times. Finally, we apply the same process as done on the MLS 2AM and 2PM zonal profiles to calculate their DW1 components.

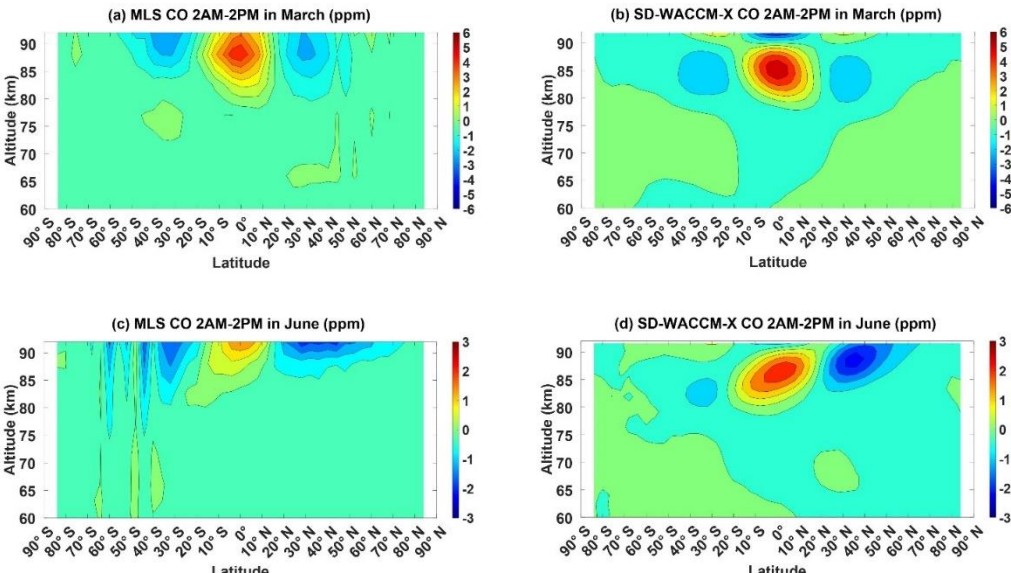

**Figure 2: Migrating Diurnal Tide component of (a) MLS CO in March equinox, (b) SD-WACCM-X CO in March equinox, (c) MLS CO in June solstice and (d) SD-WACCM-X CO in June solstice. All are in units of ppm.**

Figure 2 shows CO $\mu'$ in March equinox and in June solstice as observed by MLS and as simulated by SD-WACCM-X. For both equinox and solstice seasons, the largest MLS CO $\mu'$ and SD-WACCM-X CO $\mu'$ are found above 80 km (September equinox and December solstice shown in figure A2). Figure 2a shows that in March equinox, MLS CO $\mu'$ has peak positive perturbations of +6 ppm over the equator between 80 km and 92 km as well as peak negative perturbations of -4 ppm over the mid-latitudes. The latitude structure is consistent with the (1,1) mode in temperature; that is, peak positive anomalies over the low-latitudes and peak negative anomalies over the mid-latitudes (Forbes, 1995; Mukhartov et al, 2009). The peak negative perturbation over the southern mid-latitudes begins at around 87 km and extends above 92 km which is beyond MLS observation range. On the other hand, the peak negative perturbation over the northern mid-latitudes is located between 87 km and 92 km. Figure 2b shows that in March equinox, SD-WACCM-X CO $\mu'$ also has a peak positive perturbation of around +6 ppm over the equator but it is found between 80 km and 90 km. Above 90 km, negative perturbations of around -6 ppm are found. Over the mid-latitudes, SD-WACCM-X CO $\mu'$ has peak negative perturbations of around -4 ppm between 80 km and 90 km. Above 90 km, positive perturbations are found. SD-WACCM-X CO $\mu'$ also exhibits a latitudinal structure consistent with the (1,1) mode of temperature. However, unlike MLS, SD-WACCM-X CO $\mu'$ exhibits two local maximum (hereafter referred to as "pulse") of the (1,1) mode. The first pulse centered at around 87 km and the second pulse appears to be centered above 92 km. The pulses exhibit opposite phases of the (1,1) mode.



Figure 2c shows that in June solstice, the largest MLS CO $\mu'$ perturbations begin at around 85 km and extends beyond
92 km. MLS CO $\mu'$ has peak positive perturbations of around +2 ppm over the low-latitudes with higher values over the
northern low-latitudes than over the southern low-latitudes. Over the northern hemisphere, MLS CO $\mu'$ has peak negative
perturbations of around -3 ppm extending from latitude 30° N to latitude 50° N. Over the southern hemisphere, the
perturbations begin as negative perturbations of around -1 ppm extending from latitude 20° S to latitude 40° S. Then, it
alternates between positive and negative perturbations from latitude 40° S to latitude 60° S. Ignoring the features poleward of
latitude 40° S, the latitude structure of MLS CO $\mu'$ in June solstice is consistent with the latitude structure of temperature's
(1,1) mode distorted by the background atmosphere (Forbes, 1995; Mukhartov et al, 2009). Figure 2d shows that in June
solstice, SD-WACCM-X CO $\mu$' also has peak positive perturbations of around +3 ppm over the low-latitudes with a similar
asymmetry to that of MLS CO $\mu'$ but only extends between 80 km and 90 km. Over the northern hemisphere, SD-WACCM-
X CO $\mu'$ also has peak negative perturbations of around -3 ppm extending between latitude 30° N and 50° N but its altitudinal
extent appears to end at around 92 km. Over the southern hemisphere, SD-WACCM-X CO $\mu'$ has a negative perturbation but
it is found at lower altitudes between 80 km and 85 km. This negative perturbation extends from latitude 20° S to latitude 60°
S. SD-WACCM-X does not reproduce the alternating positive and negative perturbations from latitude 40° S to latitude 60° S
that MLS observes. Above 85 km, SD-WACCM-X CO $\mu'$ has a positive perturbation of around +0.5 ppm that also extends
from latitudes 20S to latitude 60S. SD-WACCM-X CO $\mu'$ in June solstice also shows a latitudinal structure consistent with a
distorted (1,1) mode. However, unlike MLS, SD-WACCM-X CO $\mu'$ exhibits two "pulses" of the distorted (1,1) mode. This is
like the case in March equinox. The first pulse centred at around 87 km and the second pulse centred above 92 km. The pulses
have opposite phases.

**4. DW1 Component of temperature**

Although the latitude structure of MLS CO $\mu'$ and SD-WACCM-X CO $\mu'$ have similarities to the (1,1) mode of
temperature, it has never been proven that the (1,1) mode or at least the DW1 tide affects CO. To establish this, we first
characterize the DW1 component of temperature in the region and later use this to prove that the DW1 and (1,1) tide affects
CO. Figure 3 shows $T'$ in March equinox and in June solstice as observed by MLS and by SABER and as simulated by SD-
WACCM-X (September equinox and December solstice shown in figure A3). Since we are focused on relating this to CO $\mu'$,
we focus on features above 80 km where CO $\mu'$ is largest for both seasons. Figure 3a shows that in March equinox, MLS $T'$
has peak positive perturbations of +25 K over the equator between 80 km and 92 km as well as peak negative perturbations of
-15 K over the mid-latitudes. The peak negative perturbation over the southern mid-latitudes begins at around 85 km and
appears to extend above 92 km. On the other hand, the peak negative perturbation over the northern mid-latitudes is found
between 80 km and 90 km. Figure 3c shows SABER $T'$ also in March equinox. SABER $T'$ has peak positive perturbations of
around 30K over the equator which are larger than MLS $T'$'s. This difference may be attributed to aliasing of other tides on
MLS $T'$. SABER $T'$'s peak negative perturbations over the northern and southern mid-latitudes are both found between 80 km





and 90 km unlike MLS $T'$. This difference may be attributed to the uneven sampling of MLS over the middle to high latitudes. Both MLS $T'$ and SABER $T'$ exhibit features consistent with the (1,1) mode although there are clear differences in terms of their structure's hemispheric symmetry (Forbes, 1995; Mukhartov et al, 2009). MLS $T'$'s (1,1) mode appears tilted upward because its southern mid-latitude peak appears higher than its northern mid-latitude peak. SABER $T'$'s (1,1) mode's mid-

latitude peaks occur in almost the same altitude, but the northern mid-latitude amplitudes are larger than the southern mid-latitude amplitudes. Figure 3e shows that in March equinox, SD-WACCM-X $T'$ also has a peak positive perturbation of around +25 K over the equator but it is found between 80 km and 90 km. Above 90 km, negative perturbations reaching -30 K is found over the equator. Over the mid-latitudes, SD-WACCM-X $\mu'$ has peak negative perturbations of around -10 K between 80 km and 90 km. Above 90 km, positive perturbations are found over the mid-latitudes. Overall, SD-WACCM-X $T'$ shows two

pulses of the (1,1) mode. The first pulse is centred at around 87 km and the second pulse is centred above 92 km. The pulses have opposite phases.

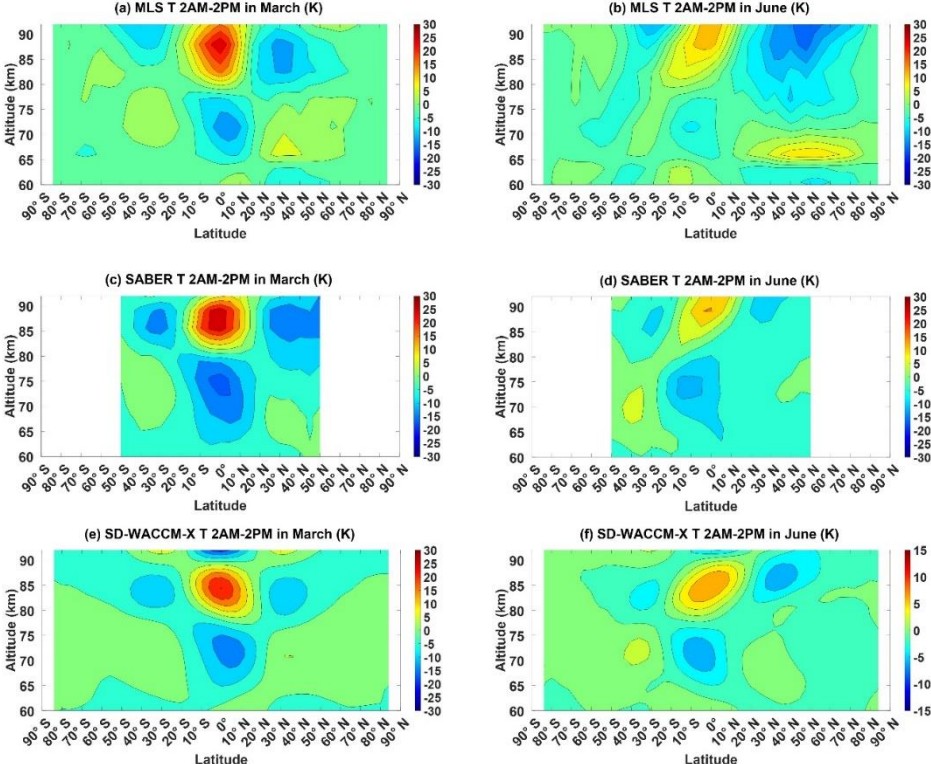

**Figure 3: Migrating Diurnal Tide component of (a) MLS temperature in March equinox, (b) MLS temperature in June solstice, (c) SABER temperature in March equinox, (d) SABER temperature in June solstice, (e) SD-WACCM-X temperature in March**

**equinox and (f) SD-WACCM-X temperature in June solstice. All are in units of K.**

In March equinox, MLS $T'$ and SABER $T'$ observe only one pulse of the (1,1) mode between 80 km and 92 km while SD-WACCM-X $T'$ simulates almost two pulses. This may be attributed to the model inaccurately simulating DW1's altitudinal





variations. On the other hand, MLS $T'$ and SABER $T'$ are different over the mid-latitudes. This shows that the differences between MLS $T'$ and SABER $T'$ over the mid-latitudes may be attributed to aliasing of other tidal components into MLS $T'$.

Figure 3b shows that in June solstice, MLS $T'$ has peak positive perturbations above 80 km over the low-latitudes with higher values over the northern low-latitudes than over the southern low-latitudes. Peak positive perturbations reach +15 K. Over the summer hemisphere, MLS $T'$ has peak negative perturbations of around -20 K extending from latitude 30° N to latitude 50° N. Over the winter hemisphere, there is a region of negative perturbations of around -15 K extending from latitude 20° S to latitude 40° S. Then, positive perturbations of around 5K are found poleward of latitude 40° S. These features are

consistent with the presence of a (1,1) mode distorted by the winter residual circulation (Forbes, 1995; McLandress, 1997; Mukhartov et al, 2009). Figure 3d shows SABER $T'$ also in June solstice. SABER $T'$ also has peak positive perturbations over the low latitudes with higher values over the northern low-latitudes than over the southern low-latitudes. Peak positive perturbations also reach +15 K. Over the summer hemisphere, SABER $T'$ also has peak negative perturbations extending from latitude 30N to latitude 50N but the magnitude is only around -7 K. Over the winter hemisphere, SABER $T'$ has peak negative

perturbations of around -7K from latitude 20S to latitude 50S. These differences between MLS $T'$ and SABER $T'$ over the mid-latitudes may be a result of MLS inadequate sampling causing significant aliasing from other tides. Our approach in calculating the DW1 component with MLS is susceptible to aliasing from the semidiurnal tide (Oberheide et al, 2003). In solstice, the migrating semidiurnal tide as well as several non-migrating semidiurnal tides are known to be significant over the winter mid-latitudes (Zhang et al, 2006). These may all contribute to the aliasing in MLS $T'$. Like MLS $T'$, these features are

also consistent with the presence of a distorted (1,1) mode. Figure 3f shows that in June solstice, SD-WACCM-X $T'$ also has peak positive perturbations over the low-latitudes with larger values over the northern low-latitudes than over the southern low-latitudes but only extends between 80 km and 90 km. Peak positive perturbations over the low-latitudes only reach +10 K. Above 90 km, negative perturbations are found over the low-latitudes. Over the summer hemisphere, SD-WACCM-X $T'$ also has peak negative perturbations of around -5 K extending between latitude 30° N and 50° N but its altitudinal extent ends

at around 90 km. Above 90 km, positive perturbations appear. Over the winter hemisphere, SD-WACCM-X $T'$ has a negative perturbation but it is found at lower altitudes between 80 km and 85 km. This negative perturbation extends from latitude 20° S to latitude 60° S. Above 85 km, SD-WACCM-X $T'$ over the winter hemisphere has a positive perturbation of around +3 K that also extends from latitudes 20° S to latitude 60° S. Unlike MLS $T'$ and SABER $T'$, SD-WACCM-X $T'$ shows two pulses of the distorted (1,1) mode above 80 km.

In June solstice, MLS $T'$ and SABER $T'$ observe only one pulse of the distorted (1,1) mode between 80 km and 92 km while SD-WACCM-X $T'$ simulates almost two pulses. This is like the case in March equinox. This indicates that the model's inaccuracies in simulating DW1's altitudinal variations occur in all seasons.





## 5. Physical Mechanisms of CO DW1

Figures 2 and 3 clearly show that the latitude-altitude structure of CO $\mu'$ and $T'$ are remarkably similar when looking at either MLS observations or SD-WACCM-X simulations. Since the latitude-altitude structure of observed or simulated $T'$ is already known to be driven by DW1, figures 2 and 3 suggest that the latitude-altitude structure of CO $\mu'$ may also be driven by DW1. However, the mechanisms of how DW1 can affect CO $\mu'$ have never been determined. In this section, we present the physical mechanisms of how DW1 affects CO. This section is divided into two sub-sections. One sub-section is about the physical mechanisms during March equinox while the other sub-section is about the mechanisms during June solstice.

### 5.1 March Equinox CO DW1

To determine the physical mechanisms, we took a two-step approach. Step 1 is a tendency analysis involving the continuity equation given by:

$$\frac{\partial \mu}{\partial t} + \left(\frac{u}{a \cos \phi}\right)\frac{\partial \mu}{\partial \lambda} + \left(\frac{v}{a}\right)\frac{\partial \mu}{\partial \phi} + w \frac{\partial \mu}{\partial z} - \frac{1}{\rho_0}\frac{\partial}{\partial z}\left(\rho_0 K_{zz}\frac{\partial \mu}{\partial z}\right) - \frac{1}{\rho_0}\frac{\partial}{\partial z}\left(\rho_0 D_\mu \frac{\partial \mu}{\partial z}\right) + \frac{1}{\rho_0}\frac{\partial}{\partial z}\left(\rho_0 \mu w_D\right) = P \qquad (2)$$

where $\mu$ is CO volume mixing ratio, $t$ is time, $\phi$ is latitude, $\lambda$ is longitude and $z$ is geopotential height. The variables $u$, $v$ and $w$ are the neutral zonal, meridional and vertical winds, respectively; $K_{zz}$ is the eddy diffusion coefficient; $D_\mu$ is the molecular diffusion coefficient and $w_D$ is its corresponding diffusive separation velocity; $P$ is the chemical production rate; $\rho_0$ is the atmospheric neutral density; and $a$ is the radius of the Earth which is $6.37 \times 10^6$ m. Note that the eddy diffusion coefficient is calculated from a gravity wave parameterization (Richter et al, 2010; Garcia et al, 2017). The molecular diffusion coefficient is as defined in Smith et al (2011). 2D least-squares fit is used to calculate the DW1 component of each term in equation 2. This determines the contributions of zonal advection, meridional advection, vertical advection, eddy diffusion, molecular diffusion and photochemical production to SD-WACCM-X $CO$ $\mu'$'s DW1 component. Comparing these terms will determine the main processes behind SD-WACCM-X CO $\mu'$. Salinas et al (2020) recently used this to determine the mechanisms of lower thermospheric carbon dioxide's ($CO_2$) local-time variations. The result of this analysis gives us SD-WACCM-X's suggested mechanisms for $\mu'$. This first method is clearly only applicable with model outputs because some of the required parameters cannot currently be observed.

Figure 4a, 4b and B1 shows the results of our tendency analysis for March equinox season. Figure 4a shows the DW1 amplitude of the time-derivative term in the continuity equation. It can be characterized by a primary equatorial peak of around 30 ppm/day between 85 km and 92 km as well as secondary mid-latitude peaks of around 12 ppm/day between similar altitudes. This latitude structure is consistent with that of the (1,1) mode. Figure 4b shows the DW1 amplitude of the vertical advection term in the continuity equation. It exhibits similar magnitude and latitude-altitude profile to figure 4a. Figure B1 shows the DW1 amplitudes of the other terms in the continuity equation. These figures show that in SD-WACCM-X, vertical advection in March equinox has the closest magnitude and latitude-altitude structure to the time-derivative term. Chemical production





has a higher magnitude than vertical advection but because, as mentioned earlier, the chemical production timescale is slower
than dynamical timescales, it's latitude-altitude structure isn't similar with the time-derivative term.

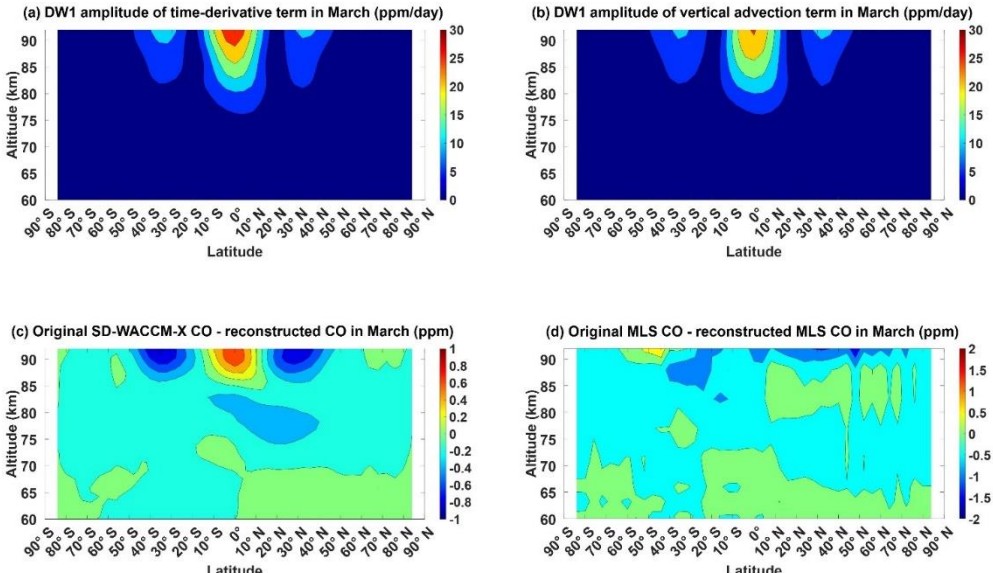

**Figure 4: Migrating Diurnal Tide component in March equinox of (a) CO's time-derivative term and (b) CO's vertical advection term. (c) Difference between SD-WACCM-X CO's DW1 component and SD-WACCM-X CO's DW1 component reconstructed using**
**adiabatic displacement method. (d) Difference between MLS CO's DW1 component and MLS CO's DW1 component reconstructed using adiabatic displacement method. Units are specified in the plots.**

Now, we need to determine if the same mechanism holds for the observations. This is step 2. A tendency analysis is
still currently impossible with observations because we do not have observations of all the needed parameters. However, we
can use a method called the adiabatic displacement method to quantify the contributions of vertical advection on CO $\mu'$ as
observed by MLS and as simulated by SD-WACCM-X. The adiabatic displacement method involves calculating the
perturbation on a tracer $\mu_w'$ due to any tide-induced vertical advection using the equation (Eckermann et al, 1998):

$$\mu_w' = \frac{T'}{S} \frac{\partial \bar{\mu}}{\partial z} \qquad (3)$$

$T'$ is the tidal perturbation of temperature. $S = \left(\frac{\partial T}{\partial z} + \frac{g}{c_p}\right)$ is static stability. $g$ is the acceleration due to gravity ($9.8\ m/s^2$). $c_p$

is the heat capacity of air at constant pressure ($1004\ J\ K^{-1}\ kg$). $\frac{\partial \bar{\mu}}{\partial z}$ is the vertical gradient of a tracer's zonal-mean profile.
Comparing CO $\mu'$ and CO $\mu_w'$ will determine how much of CO $\mu'$ is driven by vertical advection. If CO $\mu'$ and CO $\mu_w'$ are
similar, then we can argue that vertical advection does primarily drive CO $\mu'$. Applying this method on the model outputs will
assess the consistency of this method and the previous method with regards the role of vertical advection in $\mu'$. Forms of this
method have already been applied on the analysis of the tidal or local-time variations of other tracers (Akmaev et al, 1980;
Angelats I Coll and Forbes, 1998; Marsh et al, 1999; Shepherd et al, 1995; Shepherd et al, 1997; Ward, 1999; Zhang et al,
1998; Marsh and Russell, 2000; Oberheide and Forbes, 2008; Smith et al, 2010; Marsh, et al, 2011; Salinas et al, 2022)





Apart from proving that vertical advection primarily drives CO $\mu'$, equation 3 will also explain why the latitude and altitude structure of a tracer's DW1 component and temperature's DW1 component may be correlated. Equation 2 indicates that if vertical advection does primarily drive a tracer's DW1 component and if the vertical gradient of a tracer's daily-mean

zonal-mean component is positive (the gradient increases with height), CO $\mu'$ and $T'$ are correlated. On the other hand, if vertical advection does primarily drive a tracer's DW1 component and if the vertical gradient of a tracer's daily-mean zonal-mean component is negative (the gradient decreases with height), CO $\mu'$ and $T'$ are anti-correlated.

Equation 3 will also indicate whether the DW1 tide affects a tracer via net upwelling or net downwelling. We define net upwelling as either the absolute vertical wind velocity being positive or the DW1-induced change in vertical wind velocity

causes the value to be more positive. We define net downwelling as the opposite of net upwelling. The reason we use these terms is because we note that our analysis won't know for certain what the exact condition of vertical wind is unless we have vertical wind observations. However, we can deduce possible vertical wind conditions. Equation 3 indicates that if vertical advection does primarily drive a tracer's DW1 component and if the vertical gradient of a tracer's daily-mean zonal-mean component is positive, positive perturbations denote a net downwelling while negative perturbations denote a net upwelling.

On the other hand, if vertical advection does primarily drive a tracer's DW1 component and if the vertical gradient of a tracer's daily-mean zonal-mean component is negative, positive perturbations denote a net downwelling while negative perturbations denote net upwelling.

Figure 4c shows the differences between SD-WACCM-X CO $\mu'$ and SD-WACCM-X CO $\mu'_w$. The differences are within $\pm0.6$ ppm. Peak positive difference of around +0.6 ppm is found over the equator while peak negative difference of

around -0.6 ppm is found over the mid-latitudes. These differences are an order of magnitude lower from the correct values which indicates that SD-WACCM-X CO $\mu'_w$ is very similar to SD-WACCM-X CO $\mu'$. Thus, figure 4a, 4b and 4c indicate that in the model, CO $\mu'$ and CO $\mu'_w$ are very similar in March equinox. This explains the good correlation between the latitude structure of SD-WACCM-X CO $\mu'$ and SD-WACCM-X $T'$ in March equinox. We now show MLS CO $\mu' - \mu'_w$ in figure 4d. Figure 4d shows that the largest differences are above 90 km. Below 90 km, the average difference is around -0.5 ppm. This

also indicates good similarity in March equinox between MLS CO $\mu'$ and MLS CO $\mu'_w$. This explains the good correlation between the latitude structure of MLS CO $\mu'$ and MLS $T'$ in March equinox. For both MLS CO $\mu'$ and SD-WACCM-X CO $\mu'$, figures 4c and 4d indicate that the positive perturbations are driven by a relative downwelling due to the DW1 tide while the negative perturbations are driven by a relative upwelling.

## 5.2 June Solstice CO DW1

We now determine the physical mechanisms behind CO $\mu'$ in June solstice. We first show the results of the tendency analysis in SD-WACCM-X. Figure 5a shows the DW1 amplitude of the time-derivative term in the continuity equation. It can be characterized by a primary low-latitude peak of around 8-12 ppm/day between 85 km and 95 km. Amplitudes are larger over the northern low-latitude than over the southern low-latitude. Mid-latitude peaks are also present but the northern mid-





latitude peak of around 7 ppm/day is larger than the southern mid-latitude peak of around 3 ppm/day. Figure 5b shows the

DW1 amplitude of the vertical advection term in the continuity equation. It shows similar magnitude and latitude-altitude profile to figure 5a. Figure B2 shows the DW1 amplitudes of the other terms in the continuity equation. These figures show that in SD-WACCM-X, vertical advection in June solstice has the closest magnitude and latitude-altitude structure to the time-derivative term. Like March equinox, the chemical production also has a higher magnitude than vertical advection but it's latitude-altitude structure also isn't similar with the time-derivative term.

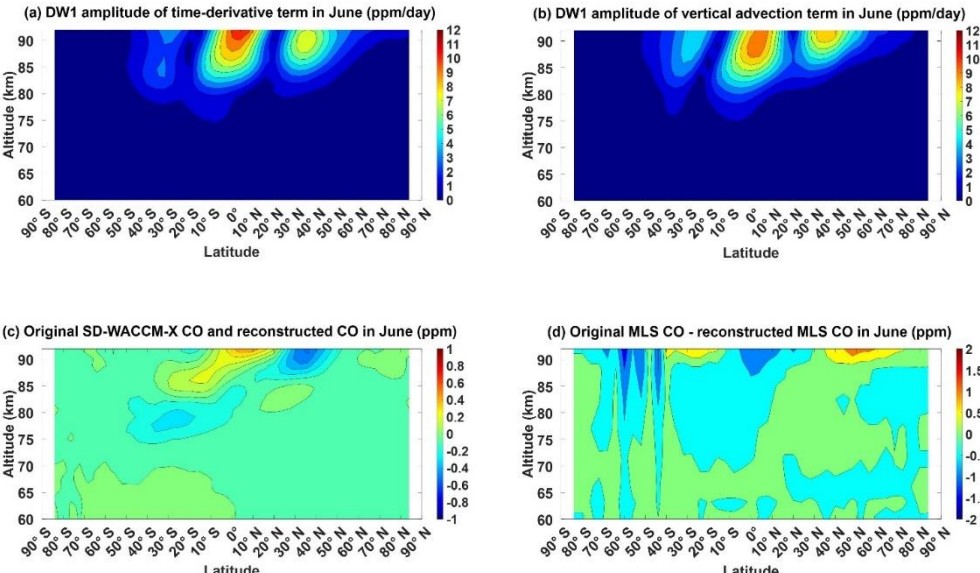

Figure 5: Same as figure 4 but for June solstice.

We now also use the adiabatic displacement method to quantify how much CO changes due to DW1-induced vertical advection in June solstice. Figure 5c shows the differences between SD-WACCM-X CO $\mu'$ and SD-WACCM-X CO $\mu'_w$ for

June solstice. The differences are within $\pm 0.5$ ppm. Peak positive difference of around +0.5 ppm is found over the equator above 90 km while peak negative difference of around -0.4 ppm is found over the northern mid-latitudes. These differences are also an order of magnitude lower from the correct values which indicates that SD-WACCM-X CO $\mu'_w$ is very similar to SD-WACCM-X CO $\mu'$. Thus, figure 5a, 5b and 5c indicate that in the model, CO $\mu'$ and CO $\mu'_w$ are remarkably similar in June solstice. These figures indicate that for SD-WACCM-X in June solstice, the positive perturbations are driven by a relative

downwelling due to the DW1 tide while the negative perturbations are driven by a relative upwelling.

We now show June solstice MLS CO $\mu' - \mu'_w$ in figure 5d. Figure 5d shows that the largest differences between latitudes 30S and 60N are above 90 km. Below 90 km, the average difference is around -0.5 ppm. This also indicates good similarity between MLS CO $\mu'$ and MLS CO $\mu'_w$ in June solstice for regions less than 90 km between latitudes 30° S and 60° N. On the other hand, between latitudes 30° S and 60° S, largest differences are found between 80 km and 92 km. This also

indicates that unlike in March equinox, MLS CO $\mu'$ and MLS CO $\mu'_w$ are not similar throughout all latitudes in June solstice.





They are only similar between latitudes 30° S and 60° N. These figures indicate that for MLS in June solstice, not all positive perturbations are driven by a relative downwelling and not all negative perturbations are driven by a relative upwelling.

## 6. (1,1) Hough Mode component's seasonal and interannual variability

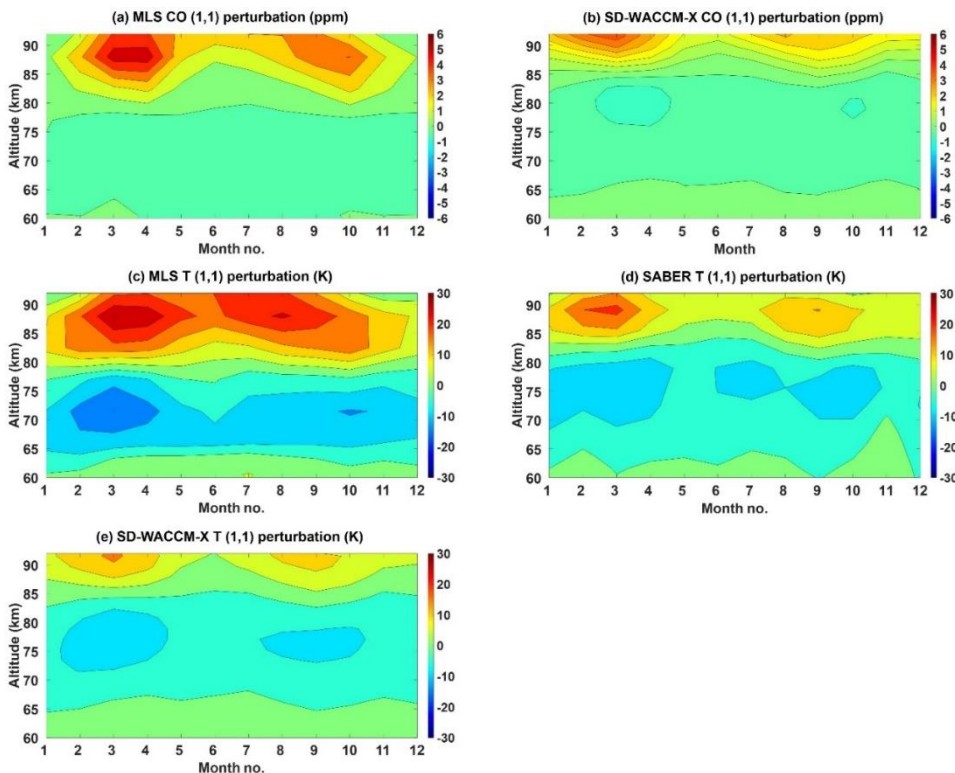

**Figure 6: Seasonality of the (1,1) component of (a) MLS CO, (b) SD-WACCM-X CO, (c) MLS temperature, (d) SABER temperature and (e) SD-WACCM-X temperature. Units are specified in the plots.**

The previous sections have shown that for both observations and simulations, CO $\mu'$ is predominantly very similar with CO $\mu'_w$. It is also shown that CO $\mu'$'s latitude-altitude structure appears predominantly comprised of the (1,1) mode. In this section, we probe deeper into the extent to which vertical advection drives CO $\mu'$ by now focusing on determining vertical advection's impact on CO's (1,1) mode across seasonal and interannual timescales. To calculate the (1,1) mode of CO $\mu'$ (hereafter referred to as CO $h'_\mu$), we regress the latitude profiles at each altitude in CO $\mu'$ with the (1,1) Hough mode profile (Forbes, 1995).

### 6.1 (1,1) Hough Mode component's seasonality

This sub-section will determine the seasonal and interannual variability of the (1,1) mode of CO. To explain it, we will first compare the (1,1) mode of CO with the (1,1) mode of temperature ($h'_T$). Then, we will determine the role of vertical





advection by regressing the latitude profiles at each altitude of CO $\mu'_w$ (presented in section 5) with the (1,1) Hough function profile. The corresponding regression coefficients will be denoted CO $h'_w$. Figure 6 first shows the seasonality of observed and modelled CO $h'_\mu$ and $h'_T$. Figure 6a shows the seasonality of MLS CO $h'_\mu$. The largest coefficients are above 80 km, and they

are all positive regression coefficients indicating that there is a positive correlation between a latitude profile of MLS CO $\mu'$ and the (1,1) Hough mode between 80 km and 92 km. This is consistent with the features shown in figure 2a and figure 2c. Between 80 km and 90 km, the seasonality is characterized by a semi-annual oscillation with primary peak of around 4 ppm in March equinox and secondary peak of around 3 ppm in September equinox. Above 90 km, it appears as though the primary peak is moving towards June solstice. Figure 6b shows the seasonality of SD-WACCM-X CO $h'_\mu$. The largest coefficients are

above 85 km, and they are all also positive regression coefficients. Between 85 km and 95 km, the seasonality is also characterized by a semi-annual oscillation with primary peak of around 6 ppm in March equinox and secondary peak of around 5 ppm in September equinox. Figure 6a and 6b shows that SD-WACCM-X does capture the observed semi-annual oscillation of the regression coefficients. However, the altitudinal variations are different from observed. This is consistent with the differences in MLS CO $\mu'$ and SD-WACCM-X CO $\mu'$. The coefficients of SD-WACCM-X CO $h'_\mu$ are also higher than MLS.

Figure 6c shows the seasonality of MLS $h'_T$. Focusing on the coefficients above 80 km, we find that, like MLS CO, they are all positive regression coefficients indicating that there is a positive correlation between MLS T$'$ and the (1,1) Hough mode between 80 km and 92 km. Between 80 km and 90 km, it also has a semi-annual oscillation with primary peak coefficients of around 25 K during March equinox and secondary peak coefficients of around 20 K during September equinox. Above 90 km, the seasonality appears to shift into an annual oscillation with peak amplitudes in June solstice. Figure 6d shows the

seasonality of SABER $h'_T$. Between 80 km and 95 km, it also has a semi-annual oscillation with primary peak of around 20 K during March equinox and secondary peak of around 15 K during September equinox. Unlike MLS $h'_T$, its seasonality doesn't seem to change above 90 km. Figure 6e shows the seasonality of SD-WACCM-X $h'_T$. Between 80 km and 95 km, it also has a semi-annual oscillation with primary peak of around 27 K during March equinox and secondary peak of around 20 K during September equinox. Unlike MLS $h'_T$, its seasonality doesn't also seem to change above 90 km. Also, SD-WACCM-X $h'_T$ is

consistently larger than SABER $h'_T$ but its coefficients aren't too different from MLS $h'_T$.

Figures 6a and 6c showed that the seasonality of MLS CO $h'_\mu$ and MLS $h'_T$ is similar. However, MLS $h'_T$ and SABER $h'_T$ in figure 6d have differences. The seasonality of MLS $h'_T$ changes slightly above 90 km while the seasonality of SABER $h'_T$ doesn't. Since SABER $h'_T$ was estimated with full local-time sampling, figures 6a, 6c and 6d shows that the seasonality of MLS CO $h'_\mu$ may be affected by the incomplete local-time sampling of MLS. This would be consistent with the results shown

in figures 2 and 3. Figures 2 and 3 showed that MLS CO $\mu'$ and MLS T$'$ may be affected by inadequate sampling over the mid-latitudes.

Figures 6a and 6b showed MLS CO $h'_\mu$ is weaker than SD-WACCM-X CO $h'_\mu$. Figures 6d and 6e also showed that SABER $h'_T$ is weaker than SD-WACCM-X $h'_T$. These figures suggest that the weaker MLS CO $h'_\mu$ may be attributed to a





weaker realistic SABER $h'_T$ than simulated. An overestimation of SD-WACCM-X $h'_T$ indicates inaccuracies in the simulated

background atmosphere, tidal source or tidal dissipation mechanisms.

**6.2 Role of Vertical Advection on the CO (1,1) Mode across Interannual timescales**

Apart from these differences amongst the datasets, figures 6a and 6c as well as figures 6b and 6e shows that CO $h'_\mu$

and $h'_T$ in MLS observations and SD-WACCM-X simulations have similarities in their seasonality. We now check whether

this similarity is also found for all months and all years of MLS observations and SD-WACCM-X simulations. We will also

check if CO $h'_\mu$ is primarily driven by vertical advection across all months and years of observations and simulations. In this

subsection, we determine the importance of vertical advection by regressing the latitude profiles at each altitude of the

previously calculated CO $\mu'_w$ with the (1,1) Hough function profile. The corresponding regression coefficients will be denoted

CO $h'_w$.

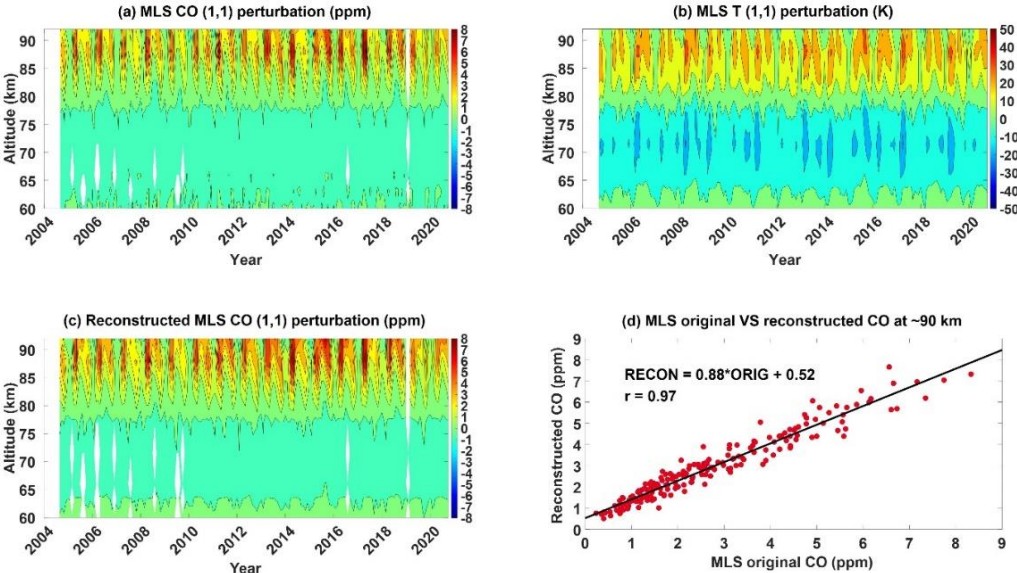

**Figure 7: (1,1) component of (a) MLS CO, (b) MLS temperature and (c) MLS CO reconstructed using the adiabatic displacement method from 2004 till 2021. (d) Scatter plot between MLS CO's (1,1) component at ~90 km and that reconstructed using the adiabatic displacement method. Units are specified in the plots.**

Figure 7a shows the MLS CO $h'_\mu$ from 2004 till 2020. Figure 7b shows MLS CO $h'_T$ from 2004 till 2020. Their overall

morphology is similar; that is, for all years, there is a semi-annual oscillation with primary peak in March equinox and

secondary peak in September equinox between 80 km and 90 km. Above 90 km, their seasonality shifts into having a primary

peak close to June solstice. Figure 7c shows the MLS CO $h'_w$. Comparing figure 7a and 7c shows that most of the seasonal

features are indeed captured. Figure 7d shows a scatter plot between CO $h'_\mu$ and CO $h'_w$ at ~90 km which is the approximate





altitude where these MLS parameters attain peak amplitudes. It shows a correlation of 0.97 indicating that the variations of $h'_\mu$

and $h'_w$ are very similar.

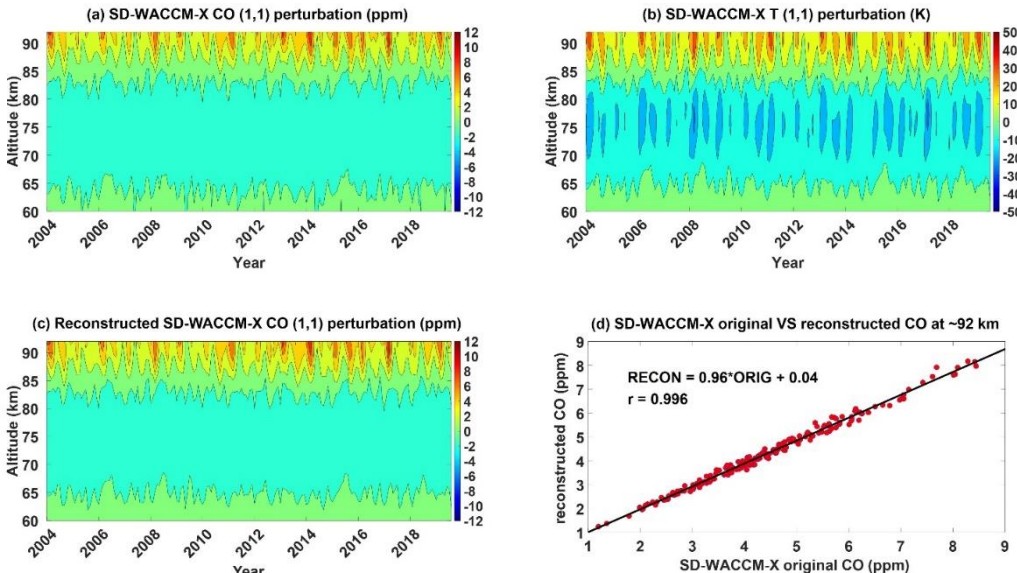

**Figure 8: (1,1) component of (a) SD-WACCM-X CO, (b) SD-WACCM-X temperature and (c) SD-WACCM-X CO reconstructed using the adiabatic displacement method from 2004 till 2021. (d) Scatter plot between SD-WACCM-X CO's (1,1) component at ~92 km and that reconstructed using the adiabatic displacement method. Units are specified in the plots.**


Figure 8a shows SD-WACCM-X CO $h'_\mu$ from 2004 till 2020. Figure 8b shows SD-WACCM-X $h'_T$ from 2004 till 2020. Their overall morphology is also similar; that is, for all years, there is a semi-annual oscillation with primary peak in March equinox and secondary peak in September equinox. We now calculate SD-WACCM-X CO $h_w'$. Figure 8c shows the SD-WACCM-X CO $h_w'$. Comparing figure 8a and 8c shows that most of the features are indeed captured. Figure 8d shows a

scatter plot between SD-WACCM-X CO $h'_\mu$ and SD-WACCM-X CO $h_w'$ at ~92 km which is the approximate altitude where these SD-WACCM-X parameters reach peak amplitudes. It shows a correlation of 0.996 indicating that the variations of CO $h'_\mu$ and CO $h_w'$ are very similar. This correlation is higher than MLS CO $h'_\mu$. We suggest that this may be due to the aliasing over the mid-latitudes in MLS CO $\mu'$. Previous studies using this adiabatic displacement method only involved the analysis of tracer observations at one instance in time (Akmaev et al, 1980; Angelats I Coll and Forbes, 1998; Marsh et al, 1999; Shepherd

et al, 1995; Shepherd et al, 1997; Ward, 1999; Zhang et al, 1998; Marsh and Russell, 2000; Oberheide and Forbes, 2008; Smith et al, 2010; Marsh, et al, 2011; Salinas et al, 2020; Salinas et al, 2022). For example, Smith et al (2010) proved that upper mesospheric SABER atomic oxygen's local time variation is primarily driven by vertical advection, but their analysis only involved the use of 7-year climatological average of vernal equinox atomic oxygen. Salinas et al (2022) recently did something very similar but for mesospheric SABER water vapor. Our work adds to these studies by presenting an approach involving the

adiabatic displacement method that involves proving the importance of vertical advection at both seasonal and interannual timescales.





## 6.3 Cross-wavelet Analysis

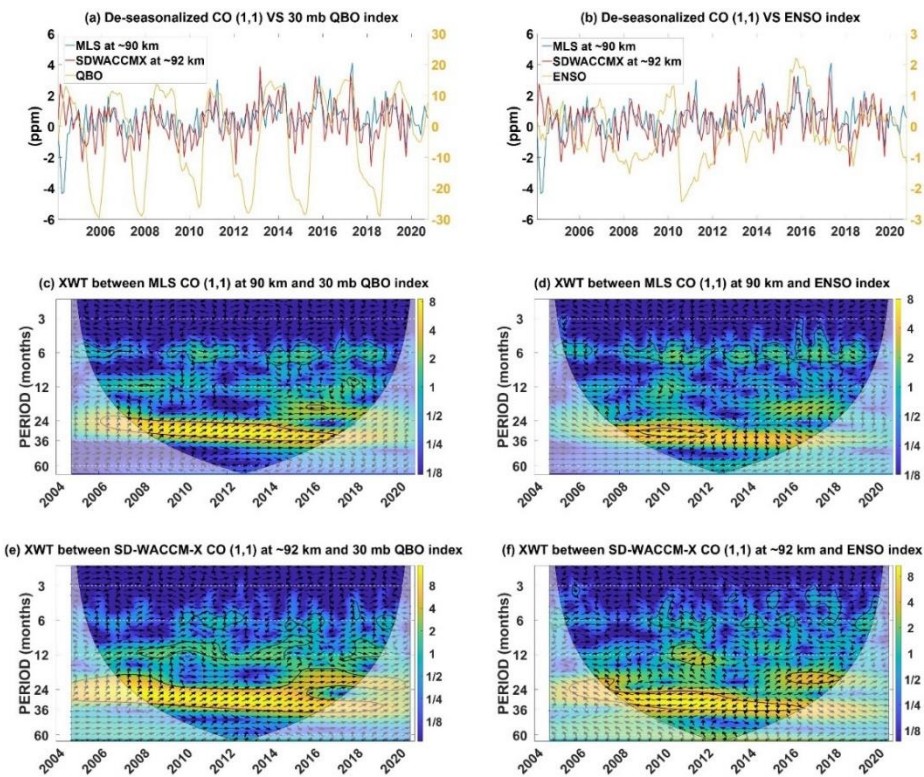

**Figure 9: (a) De-seasonalized time-series of MLS CO (1,1) at ~90 km, SD-WACCM-X CO (1,1) at ~92 km and the QBO index. (b)**
**De-seasonalized time-series of MLS CO (1,1) at ~90 km, SD-WACCM-X CO (1,1) at ~92 km and the ENSO index. (c) Cross-wavelet**
**(XWT) spectrum between MLS CO (1,1) at ~90 km and the QBO index. (d) Cross-wavelet spectrum between MLS CO (1,1) at ~90**
**km and the ENSO index. (e) Cross-wavelet spectrum between SD-WACCM-X CO (1,1) at ~92 km and the QBO index. (f) Cross-**
**wavelet spectrum between SD-WACCM-X CO (1,1) at ~92 km and the ENSO index.**

Figures 7 and 8 indicate that across interannual timescales, both observed and modelled CO $h'_\mu$ and CO $h'_w$ are highly

correlated. In this sub-section, we identify the interannual phenomena in CO $h'_\mu$ that can apparently also be seen in CO $h'_w$

because they are both highly correlated. We focus on interannual phenomena that are known to already affect the (1,1) mode

in temperature: the Quasi-biennial Oscillation and the El Nino Southern Oscillation (Lieberman, 1997; Vincent et al., 1998;

McLandress, 2002a; 2002b; Gurubaran et al, 2005; Mayr & Mengel, 2005; Liebermann et al, 2007; Wu et al, 2008; Gurubaran

et al., 2009; Mukhtarov et al, 2009; Pancheva et al, 2009; Xu et al., 2009; Pedatella et al, 2012; Pedatella et al, 2013; Gan et

al, 2014; Liu et al, 2017; Zhou et al, 2018; Kogure et al, 2021; Pramitha et al, 2021; Cen et al, 2022). We use a cross-wavelet

analysis to determine the dominant oscillations in MLS CO $h'_\mu$ or SD-WACCM-X CO $h'_\mu$ that coincide with the 30 mb QBO

index and Multi-variate El Nino Southern Oscillation index (MEI).

    Figure 9 identifies interannual phenomena found in CO $h'_\mu$. Figure 9a shows the time-series of MLS CO $h'_\mu$ at ~90

km, SD-WACCM-X CO $h'_\mu$ at ~92 km and the QBO index. Figure 9b shows the time-series of MLS CO $h'_\mu$ at ~90 km, SD-





WACCM-X CO $h'_\mu$ at ~92 km and the MEI index. Figure 9c shows the cross-wavelet spectrum between MLS CO $h'_\mu$ and the QBO index. In this and the succeeding spectra, encircled regions with the high amplitudes correspond to oscillations statistically significant in both time-series (Grinsted et al, 2004). The arrows indicate the phase relationship between the time-series. If the arrow points right, both time-series are in phase. If the arrow points left, both time-series are anti-phase. If the

arrow points upward or downward, there is a lag between the time-series. In our case, upward arrow indicates that both time-series are in phase but the QBO or ENSO index time-series peaks later than the other time-series. Downward arrow indicates that both time-series are also in phase but the QBO or ENSO index time-series peaks ahead of the other time-series. Figure 9c reveals that both MLS CO $h'_\mu$ and the QBO index have a statistically significant oscillation with periods of around 24 months. It is statistically significant between 2005 and 2018. The arrows are pointed slightly upward which indicate that MLS CO $h'_\mu$

peak slightly ahead of the QBO index. The arrows also indicate that MLS CO $h'_\mu$ increases during the westerly phase of the QBO while it decreases during the easterly phase of the QBO. Remarkably similar features are found in the cross-spectrum between MLS $h'_T$ and the QBO index (figure C1C). The QBO's impacts on the (1,1) mode of temperature is well-known (Lieberman, 1997; Vincent et al., 1998; McLandress, 2002; Mayr & Mengel, 2005; Wu et al, 2008; Gurubaran et al., 2009; Mukhtarov et al, 2009; Pancheva et al, 2009; Xu et al., 2009; Gan et al, 2014; Pramitha et al, 2021). These studies have shown

that the (1,1) mode is enhanced during the westerly phase of the QBO while it is reduced during the easterly phase of the QBO. Our works adds to these previous studies by showing that MLS CO's (1,1) mode is also affected by the QBO in the same way.

Figure 9d shows the cross-wavelet spectrum between MLS CO $h'_\mu$ and the MEI index. This spectrum reveals that both MLS CO $h'_\mu$ and the MEI index have a statistically significant oscillation of around 30 months between 2008 and 2012. This does coincide with the strong 2010 – 2011 La Nina event. The arrows are pointed almost fully to the left which indicate that

MLS CO $h'_\mu$ and ENSO are anti-correlated during this event; that is, MLS CO $h'_\mu$ increased during this La Nina event. The anti-correlation also indicates that, if we were to solely use this event as a basis, MLS CO $h'_\mu$ should decrease during El Nino events. Very similar features are found in the cross-spectrum between MLS $h'_T$ and the ENSO index (figure C1D). ENSO's impacts on the (1,1) mode of temperature is well explored (Gurubaran et al, 2005; Liebermann et al, 2007; Pedatella et al, 2012; Pedatella et al, 2013; Liu et al, 2017; Zhou et al, 2018; Kogure et al, 2021; Cen et al, 2022). These studies have shown

that the general response is that (1,1) mode is reduced during El Nino although the reduction is modulated by secondary mechanisms like gravity waves (Cen et al, 2022).

Figure 9e shows the cross-wavelet spectrum between SD-WACCM-X CO $h'_\mu$ and the QBO index. This spectrum reveals that both SD-WACCM-X CO $h'_\mu$ and the QBO index have a statistically significant oscillation with periods ranging from 20 to 30 months and the statistical significance is found throughout all years. From 2015 to 2020, both have a statistically

significant oscillation with period around 15 to 20 months. Very similar features are found in the cross-spectrum between SD-WACCM-X $h'_T$ and the QBO index (figure C1E). Like MLS CO $h'_\mu$, the arrows are pointed slightly upward which indicate that SD-WACCM-X CO $h'_\mu$ peak slightly ahead of the QBO index and that SD-WACCM-X CO $h'_\mu$ increases during the





westerly phase of the QBO while it decreases during the easterly phase of the QBO. The differences between figure 9c and 9e could suggest that the model may be overestimating the impacts of the QBO during certain periods.

Figure 9f shows the cross-wavelet spectrum between SD-WACCM-X CO $h'_\mu$ and the MEI index. This spectrum reveals that both have a statistically significant oscillation with period of around 24 to 36 months. This period of statistical significance lasts from 2006 to 2016. Unlike MLS CO $h'_\mu$, the statistical significance for SD-WACCM-X CO $h'_\mu$ includes both the 2010-2011 La Nina and the 2015-2016 El Nino event. The differences between figure 9d and 9f could suggest that the model may be overestimating the impacts of ENSO during certain periods. Very similar features are found in the cross-
spectrum between SD-WACCM-X $h'_T$ and the ENSO index (figure C1F). Like MLS $h'_\mu$, the arrows are pointed almost fully to the left between 2006 to 2013 which indicate that, for the 2010-2011 La Nina period, SD-WACCM-X CO $h'_\mu$ also increased. However, between 2013 and 2016, the arrows are pointed almost downward which indicate that, for the 2015-2016 El Nino period, SD-WACCM-X CO $h'_\mu$ increased. In addition, the arrows also indicate that ENSO peaks ahead of SD-WACCM-X CO $h'_\mu$. Most studies have found that the (1,1) mode should decrease during El Nino events. However, our results indicate that the
effect of ENSO reversed during the 2015 El Nino. Kogure et al (2021) has explained this. Their work showed that the enhanced (1,1) tide in 2015 was a result of the overlapping occurrence of an easterly QBO phase and an El Nino event. Our works adds to these previous studies by showing that MLS CO's (1,1) mode is also affected by ENSO in the same way.

**6.4 Quantifying the QBO and ENSO response using multiple linear regression analysis and a lowpass filter**

The previous sub-section found that QBO and ENSO variabilities are present in both MLS CO $h'_\mu$ and SD-WACCM-
X CO $h'_\mu$. In this section, we quantify the changes in MLS CO $h'_\mu$ or SD-WACCM-X CO $h'_\mu$ due to QBO and ENSO. We use multiple linear regression (MLR) to estimate the response of MLS CO $h'_\mu$ or SD-WACCM-X CO $h'_\mu$ to the QBO and ENSO. Finally, we use the same MLR to reconstruct the time-series with the QBO and ENSO indices and then compare this reconstruction with a lowpass filtered version of the time-series. Note that while an MLR analysis offers an estimate of the response of one parameter to another, the reconstructed time-series using these MLR coefficients constrains the fluctuations
to either be in-phase or completely anti-phase of the other parameter. It also assumes that the amplitude fluctuations are the same as that of the other time-series. For example, in the case of MLS CO $h'_\mu$ and the QBO index, the MLR reconstruction of MLS CO $h'_\mu$ can only be a time-series that is either completely in-phase with the QBO index or completely anti-phase. The overall fluctuation of the reconstruction will also only be a multiple of the QBO index time-series. With the lowpass filtered time-series, we can reconstruct the time-series that accounts for non-in-phase or non-anti-degree phase differences. The
lowpass filtered timeseries also accounts for the exact amplitude fluctuations as a function of time. Thus, in the case of MLS CO $h'_\mu$ and the QBO index, the lowpass filtered MLS CO $h'_\mu$ will reveal how dominant QBO periodicities are.

Figure 10 shows our MLR analysis and our low-pass filtering of MLS CO $h'_\mu$ and SD-WACCM-X CO $h'_\mu$ to quantify the responses of these parameters to QBO and ENSO. Figure 9 showed that the phase-relation between MLS CO $h'_\mu$ or SD-



WACCM-X CO $h'_\mu$ and the QBO index were consistent for all years. However, the phase-relation between MLS CO $h'_\mu$ or

SD-WACCM-X CO $h'_\mu$ and the QBO index changed. In this figure, we separate this analysis. Figure 10a and 10b focuses on

the QBO response while figures 10c to 10f focuses on the ENSO response.

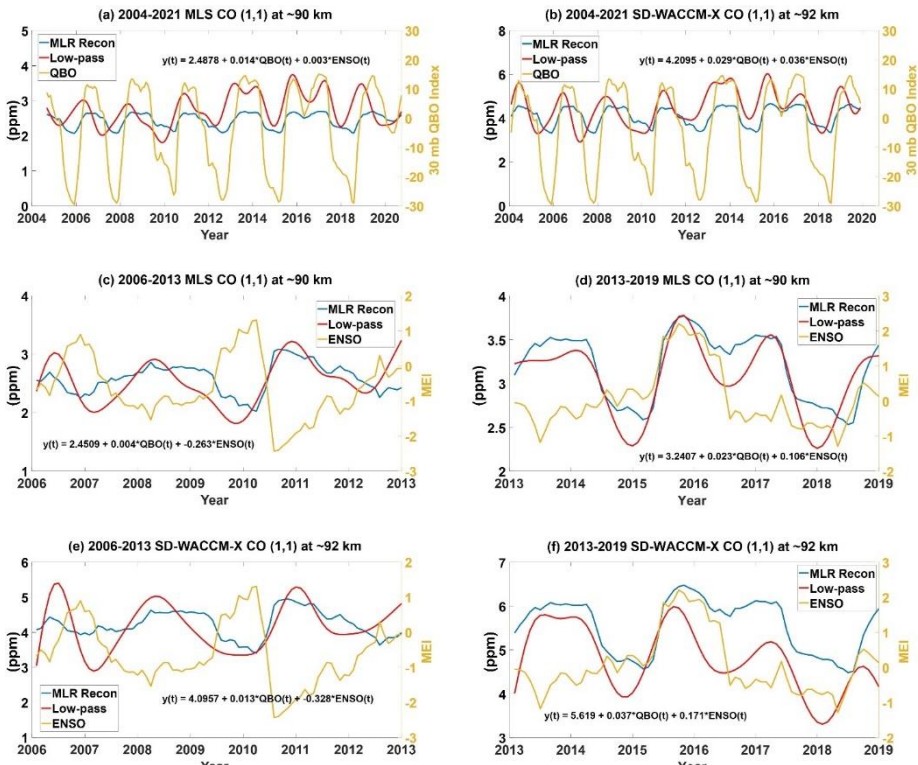

**Figure 10: (a) MLS CO (1,1) at ~90 km reconstructed using multiple linear regression (MLR), low pass filtered and the 30 mb QBO index from 2004 till 2020. (b) Same as (a) but for SD-WACCM-X CO (1,1) at ~92 km. (c) MLS CO**

**(1,1) at ~90 km reconstructed using multiple linear regression, low pass filtered and the Multi-variate ENSO index from 2006 till 2013. (d) Same as (c) but for years 2013 till 2019. (e) SD-WACCM-X CO (1,1) at ~92 km reconstructed using multiple linear regression, low pass filtered and the Multi-variate ENSO index from 2006 till 2013. (f) Same as (e) but for years 2013 till 2019. Equations within the subplots are the MLR fits.**

Figures 10a shows the de-seasonalized MLS CO $h'_\mu$ from 2004 till 2020 reconstructed using an MLR (hereafter MLR

recon MLS CO $h'_\mu$) and filtered with a 5th order low-pass filter with a cut-off period of 6-months (hereafter lowpass filtered

MLS CO $h'_\mu$). The equation within the plot shows the MLR coefficients between the de-seasonalized MLS CO $h'_\mu$ and the

QBO and ENSO indices. However, for this plot, we will ignore the coefficients for ENSO because, as mentioned above and

as will be shown later, the phase relationship changes. Both reconstructions are overplotted with the QBO index. Figure 10b

shows the same as figure 10a but for SD-WACCM-X CO $h'_\mu$. Figure 10a shows a QBO MLR coefficient of 0.014 for MLS





CO $h'_\mu$. The MLR fit has p-value = 0.002 which is less than 0.05 indicating statistical significance. With this value, MLS CO

$h'_\mu$ increases by around 0.21 ppm or a +8% variation ($\frac{100(0.21)}{2.49} = 8\%$ variation where 2.49 is the temporal mean included in

the MLR equation) when the QBO index is at the typical peak westerly value of around + 15 m/s. On the other hand, MLS CO

$h'_\mu$ decreases by around -0.42 ppm or a -16% variation when the QBO index is at the typical peak easterly value of around -30

m/s. Figure 10a also shows that the lowpass filtered MLS CO $h'_\mu$ is dominated by low frequency periodicities whose

combination yields a timeseries that is dominated by QBO-like fluctuations. The time-series looks very similar to MLR recon

MLS CO $h'_\mu$ and the QBO index. Using the standard deviation of the lowpass filtered MLS CO $h'_\mu$ as a measure of the variation,

we calculate a variation of ~0.9 ppm which is slightly higher than the 0.63 ppm variation estimated with the MLR coefficients.

Comparing the low pass filtered MLS CO $h'_\mu$ and the QBO index further shows that peak westerly phase values in the QBO

index occurs just after the local maximum values. This is consistent with the cross-wavelet spectrum arrows in figure 9c.

Figure 10b shows a QBO MLR coefficient of 0.029 for SD-WACCM-X CO $h'_\mu$. The MLR fit has p-value less than 0.0001

indicating statistical significance (p-value < 0.05). With this value, SD-WACCM-X CO $h'_\mu$ increases by around 0.44 ppm or

a +10% variation when the QBO index is at the typical peak westerly value. SD-WACCM-X CO $h'_\mu$ decreases by around 0.88

ppm or a -20% variation when the QBO index is at the typical peak easterly value. Figure 10b also shows that the lowpass

filtered SD-WACCM-X CO $h'_\mu$ is also dominated by QBO-like fluctuations that are like the MLR recon SD-WACCM-X CO

$h'_\mu$ but slightly larger in variance at around 1.52 ppm and corrected for the phase differences determined by the cross-wavelet

spectrum arrows in figure 9d.

Figures 10c shows the de-seasonalized MLS CO $h'_\mu$ from 2006 till 2013 reconstructed using an MLR and filtered

with a 5[th] order low-pass filter with a cut-off period of 6-months. Both reconstructions are overplotted with the ENSO index.

Figures 10d shows the de-seasonalized MLS CO $h'_\mu$ from 2013 till 2019 reconstructed using an MLR and filtered with a 5[th]

order low-pass filter with a cut-off period of 6-months. Both reconstructions are overplotted with the ENSO index. Figures

10e and 10f is the same as figures 10c and 10d, respectively, but for SD-WACCM-X CO $h'_\mu$.

Figure 10c focuses on the 2010-2011 La Nina event. Figure 10c shows an ENSO MLR coefficient of -0.263 for MLS

CO $h'_\mu$ between years 2006 and 2013. The MLR fit has p-value = 0.0053 indicating statistical significance (p-value < 0.05)..

With this value, MLS CO $h'_\mu$ increases by 0.526 ppm or an around +20% variation during the 2010-2011 La Nina event. Figure

10c also show that the lowpass filtered MLS CO $h'_\mu$ is dominated by periodicities whose combination yields a timeseries that

looks like the MLR recon MLS CO $h'_\mu$. The lowpass filtered timeseries also shows a clear anti-phase relation with the MEI

index. This is consistent with the cross-wavelet spectrum arrows in figure 9d. Unlike the analysis involving the QBO index in

figures 10a and 10b though, the variance of the lowpass filtered MLS CO $h'_\mu$ and the MLR recon MLS CO $h'_\mu$ in figure 10c is

very similar.

Figure 10d focuses on the 2015-2016 El Nino event. Figure 10d shows an ENSO MLR coefficient of 0.106 for MLS

CO $h'_\mu$ between years 2013 and 2019. The MLR fit has p-value = 0.0019 indicating statistical significance (p-value < 0.05).





With this value, MLS CO $h'_\mu$ increased by 0.212 ppm or around +6% variation for during the El Nino event. The coefficients shown in figure 10c and 10d are consistent with the results of the cross-wavelet analysis.

Figure 10e shows an ENSO MLR coefficient of -0.328 for SD-WACCM-X CO $h'_\mu$ between years 2006 and 2013. The MLR fit has p-value = 0.0009 indicating statistical significance (p-value < 0.05). With this value, SD-WACCM-X CO $h'_\mu$ increases by around 0.656 ppm or an around +16% variation during the 2010-2011 La Nina event. Figure 10f shows an ENSO MLR coefficient of 0.171 for SD-WACCM-X CO $h'_\mu$ between years 2013 and 2019. The MLR fit has p-value = 0.0001 indicating statistical significance (p-value < 0.05). With this value, SD-WACCM-X CO $h'_\mu$ increases by around 0.342 ppm or around +6% variation during the 2015-2016 El Nino event. Apart from the coefficients, the qualitative descriptions in figure 10e and 10f are very similar to that in figures 10c and 10d, respectively.

## 7. Summary and Conclusions

This work uses 17 years of CO observations provided by the Microwave Limb Sounder (MLS) on-board the Aura satellite to analyse the seasonal and interannual variability of the DW1 component of upper mesospheric CO. These were then compared to simulations by the Specified Dynamics – Whole Atmosphere Community Climate Model with Ionosphere/Thermosphere eXtension (SD-WACCM-X). Our results showed that the largest MLS CO $\mu'$ and SD-WACCM-X CO $\mu'$ are above 80 km. For MLS CO $\mu'$, its latitude structure in March equinox above 80 km resembles that of the (1,1) mode although there is an interhemispheric asymmetry with the location of their mid-latitude peaks. The southern mid-latitude peaks are located higher than the northern mid-latitude peaks. On the other hand, the latitude structure of MLS CO $\mu'$ in June solstice above 80 km resembles that of the distorted (1,1) mode. For SD-WACCM-X CO $\mu'$, it's latitude structure in March equinox above 80 km also resembles that of the (1,1) mode but there is negligible interhemispheric asymmetry with the location of their mid-latitude peaks. Also, SD-WACCM-X simulates two pulses of this (1,1) mode feature between 80 km and 95 km while MLS observes only one pulse. SD-WACCM-X CO $\mu'$ in June solstice also resembles that of the distorted (1,1) mode but SD-WACCM-X simulates two pulses of this mode.

To explain MLS CO $\mu'$ and SD-WACCM-X CO $\mu'$, we first looked at MLS T$'$, SABER T$'$ and SD-WACCM-X T$'$. MLS T$'$, SABER T$'$ and SD-WACCM-X T$'$ also show the (1,1) mode in March equinox and the distorted (1,1) mode in June solstice. However, the (1,1) mode in March equinox for MLS T$'$ shows more interhemispheric asymmetry in terms of the locations of the mid-latitude peaks. Also, SD-WACCM-X T$'$ shows two pulses of the (1,1) mode and distorted (1,1) mode.

Comparing MLS CO $\mu'$ and MLS T$'$ in March equinox shows that MLS CO $\mu'$ and MLS T$'$ are closely correlated. However, MLS CO $\mu'$ and MLS T$'$ in June solstice shows that MLS CO $\mu'$ and MLS T$'$ are only closely correlated between latitudes 30° S and 60° N. On the other hand, SD-WACCM-X CO $\mu'$ and SD-WACCM-X T$'$ are closely correlated for both seasons and for all latitudes. These gave hints that the mechanisms driving CO $\mu'$ may indeed be related to the mechanisms behind T$'$.





To determine what drives CO $\mu'$ and how it relates to T′, we first did a tendency analysis involving the continuity equation. Our tendency analysis revealed that, in SD-WACCM-X, vertical advection in March equinox has the closest magnitude and latitude-altitude structure to the time-derivative term. We then determined if the same mechanism holds for the observations by using the adiabatic displacement method. Our adiabatic displacement method determined that for March equinox, CO $\mu'$ and CO $\mu'_w$ in observations and simulations were very similar.

For June solstice, our tendency analysis showed that in SD-WACCM-X, vertical advection also has the closest magnitude and latitude-altitude structure to the time-derivative term. Our adiabatic displacement method then determined that for the model, CO $\mu'$ and CO $\mu'_w$ are very similar. On the other hand, for the observations, CO $\mu'$ and CO $\mu'_w$ are only similar between latitudes 30° S and 60° N.

After comparing CO $\mu'$ and CO $\mu'_w$ in observations and simulations, we probed deeper into CO's (1,1) mode. Our results showed that for seasonal and interannual timescales, the observed and simulated CO $h'_\mu$ and CO $h'_w$ are highly correlated with correlation coefficients of at least 0.97.

Finally, we characterized the interannual variability present in CO $\mu'$. A cross-wavelet between MLS CO $h'_\mu$ at ~90 km and the QBO index shows that MLS CO $h'_\mu$ and QBO index both have statistically significant oscillations with periods of around ~24 months between years 2005 and 2018. The MLR analysis and lowpass filtering further indicate that MLS CO $h'_\mu$ is enhanced by around 8% during the westerly phase of the QBO and is reduced by around 16% during the easterly phase of the QBO. On the other hand, a cross-wavelet between SD-WACCM-X CO $h'_\mu$ at ~92 km and the QBO index shows that SD-WACCM-X CO $h'_\mu$ and QBO index both have statistically significant oscillations with periods between 20 to 30 months for all years. In addition, SD-WACCM-X CO $h'_\mu$ and QBO index both have statistically significant oscillations with period of around between 15 to 20 months from 2015 to 2020. Despite these differences in the statistically significant periods, the MLR analysis and lowpass filtering indicate that SD-WACCM-X CO $h'_\mu$ is also enhanced by around 10% during the westerly phase of the QBO and is also reduced by around 20% during the easterly phase of the QBO. These indicate that though both MLS CO $h'_\mu$ and SD-WACCM-X CO $h'_\mu$ may be affected by the QBO and the correlation between the QBO phase and CO $h'_\mu$ maybe similar, the temporal coverage of the influence differs between the observations and simulations.

A cross-wavelet between MLS CO $h'_\mu$ at ~90 km and the ENSO index shows that MLS CO $h'_\mu$ and ENSO index both have statistically significant oscillations with periods of around ~30 months between years 2008 and 2012. This coincides with the strong 2010-2011 La Nina event. The MLR analysis and lowpass filtering indicates that MLS CO $h'_\mu$ is enhanced by around 20% during this La Nina event. On the other hand, a cross-wavelet between SD-WACCM-X CO $h'_\mu$ at ~92 km and the ENSO index shows that SD-WACCM-X CO $h'_\mu$ and ENSO index both have statistically significant oscillations with periods between 24 to 36 months from 2006 till 2016. This coincides with both the strong 2010-2011 La Nina event and the strong 2015-2016 El Nino event. The MLR analysis and lowpass filtering indicates that SD-WACCM-X CO $h'_\mu$ is enhanced by around 16% during the 2010-2011 La Nina event and by around 6% during the 2015-2016 El Nino event.





From these results, we can conclude that the global structure of upper mesospheric MLS CO's DW1 component is primarily driven by DW1-induced vertical advection over all latitudes during equinox seasons and over all latitudes except the winter middle to high latitudes during solstice seasons. This could suggest that MLS CO's DW1 component over the winter middle to high latitudes may be driven by other mechanisms such as meridional advection, eddy diffusion and/or chemistry. It could also suggest that the data over the winter middle to high latitudes may be affected by inadequate sampling. On the

other hand, the global structure of upper mesospheric SD-WACCM-X CO's DW1 component is primarily driven by DW1-induced vertical advection over all latitudes for both equinox and solstice seasons. We also conclude that the dominant DW1 tidal mode in upper mesospheric MLS CO DW1 and SD-WACCM-X CO DW1 is the (1,1) mode and that for seasonal and interannual timescales, the (1,1) mode affects MLS CO DW1 and SD-WACCM-X CO DW1 primarily through vertical advection. However, MLS only observes one pulse of the (1,1) mode between 80 km and 95 km while SD-WACCM-X

simulates two pulses. This could be due to MLS' limited vertical resolution, or it could be due to inaccuracies in SD-WACCM-X simulation of the background atmosphere and/or tidal vertical propagation. In addition, we find that the interannual variability of MLS CO (1,1) and SD-WACCM-X CO (1,1) is primarily driven by the QBO and ENSO's effects on DW1-induced vertical advection. These conclusions suggest that we can use CO as a tracer for vertical advection due to the DW1 tide and the (1,1) mode on seasonal and interannual timescales.


**Appendix A: Daily-mean zonal-mean CO as well as the CO DW1 and temperature DW1 in September equinox and December solstice**

Figure A1 shows the CO $\bar{\mu}$ averaged for all September equinox and for all December solstice as observed by MLS and as simulated by SD-WACCM-X. Figure A2 shows CO $\mu'$ in September equinox and in December solstice as observed by

MLS and as simulated by SD-WACCM-X. Figure A3 shows $T'$ in September equinox and in December solstice as observed by MLS and SABER and as simulated by SD-WACCM-X. The similarities and differences of these parameters between September equinox and December solstice are the same as those in the comparison of these parameters between March equinox and June solstice.





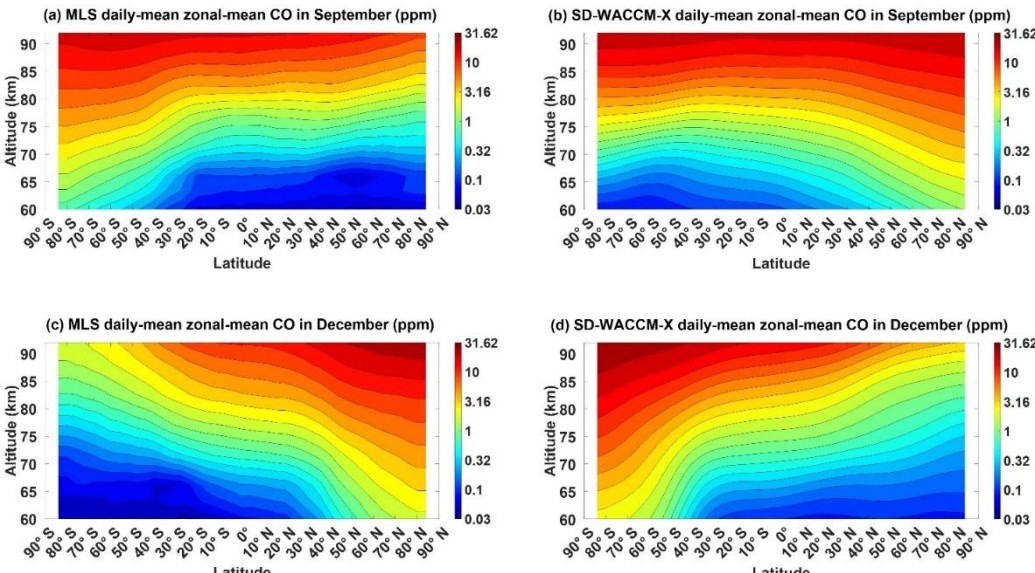


**Figure A1: Daily-mean zonal-mean component of (a) MLS CO in September equinox, (b) SD-WACCM-X CO in September equinox, (c) MLS CO in December solstice and (d) SD-WACCM-X CO in December solstice. All are in units of ppm.**

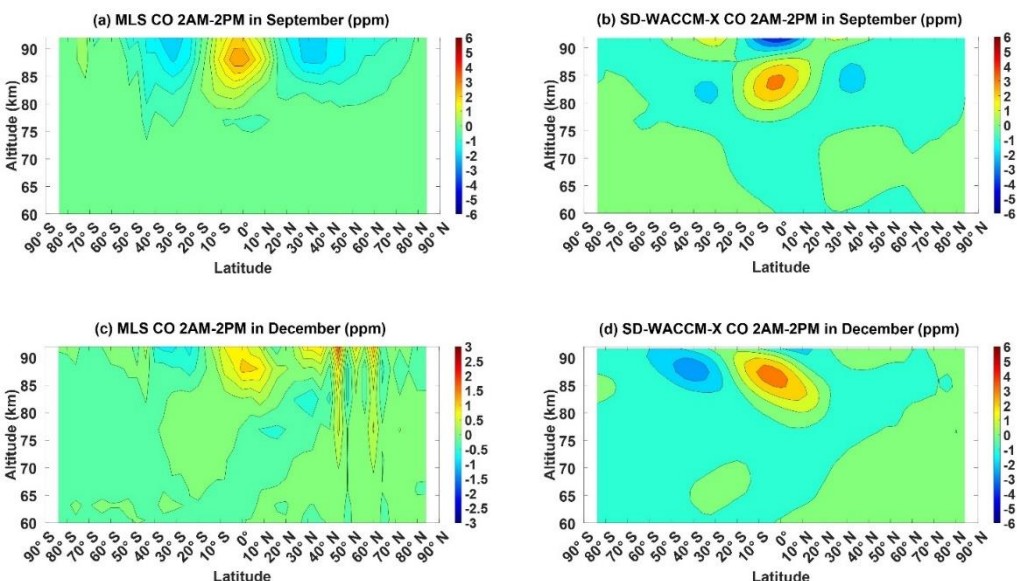


**Figure A2: Migrating Diurnal Tide component of (a) MLS CO in September equinox, (b) SD-WACCM-X CO in September equinox, (c) MLS CO in December solstice and (d) SD-WACCM-X CO in December solstice. All are in units of ppm.**





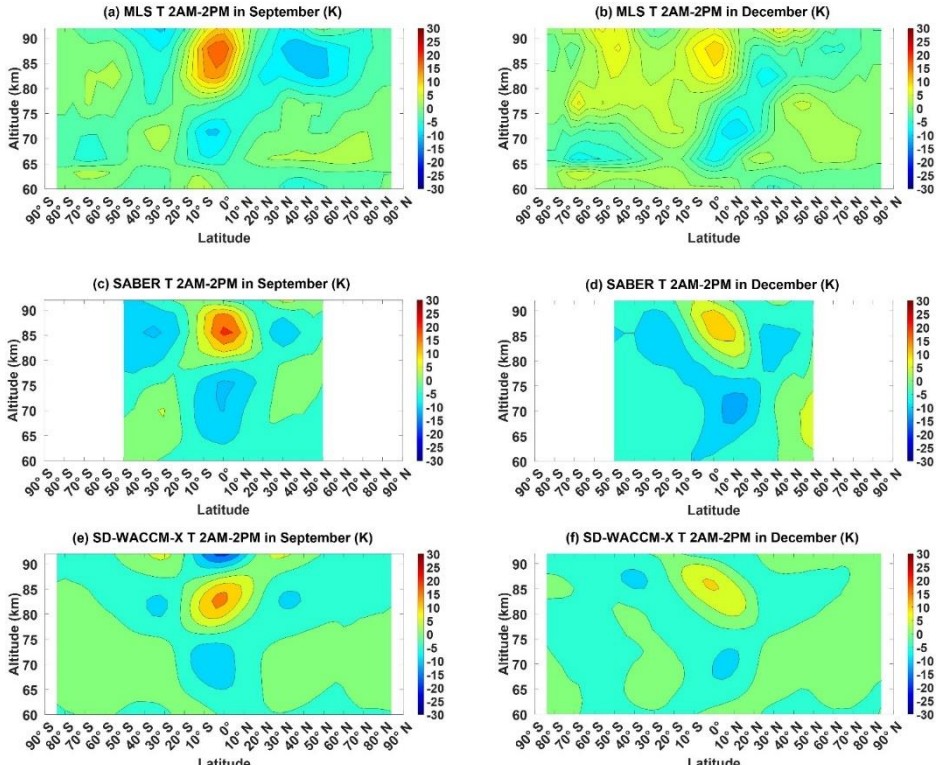

**Figure A3: Migrating Diurnal Tide component of (a) MLS temperature in September equinox, (b) MLS temperature in December**
**solstice, (c) SABER temperature in September equinox, (d) SABER temperature in December solstice, (e) SD-WACCM-X temperature in September equinox and (f) SD-WACCM-X temperature in December solstice. All are in units of K.**

**Appendix B: Tendency Analysis terms**

Figure B1 shows the chemical production term, meridional advection term, eddy diffusion term and molecular
diffusion term of CO for March equinox. Figure B1A shows the chemical production term peaking to around 2 ppm/day over
the equator above 90 km. Figure B1B shows the meridional advection term peaking to around 4 ppm/day over the mid-latitudes
above 90 km. Figure B1C shows the eddy diffusion term peaking to around 0.6 ppm/day over the low-latitudes above 90 km.
Figure B1D shows the molecular diffusion term peaking to around 0.2 ppm/days also over the low-latitudes above 90 km.
These values are all clearly significantly lower than the vertical advection term in figure 4B.






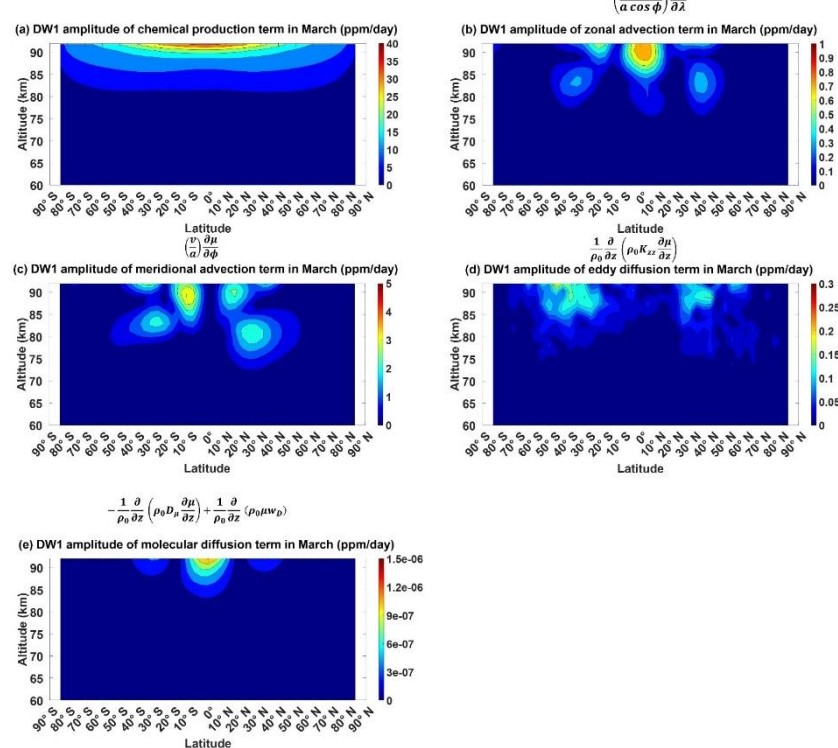

**Figure B1: Migrating Diurnal Tide component in March equinox of (a) CO's chemical production term, (b) CO's zonal advection term, (c) CO's meridional advection term, (d) CO's eddy diffusion term and (e) CO's molecular diffusion term. All are in units of ppm/day.**


Figure B2 shows the chemical production term, meridional advection term, eddy diffusion term and molecular diffusion term of CO for June solstice. Figure B2A shows the chemical production term peaking to around 4 ppm/day also over northern high latitudes above 90 km. Figure B2B shows the meridional advection term peaking to around 4 ppm/day over the mid-latitudes above 80 km. Figure B2C shows the eddy diffusion term peaking to around 0.3 ppm/day over the northern low-latitudes above

90 km. Figure B2D shows the molecular diffusion term peaking to around 0.1 ppm/days also over the northern low-latitudes above 90 km. Unlike March equinox, there are regions where the chemical production term and meridional advection term are comparable to the vertical advection term. For the chemical production term, its values aren't too far from the vertical advection term over the northern high-latitudes above 90 km. For the meridional advection term, its values aren't too far from vertical advection term over the northern mid-latitudes between 80 km and 90 km. These terms could function as secondary

mechanisms over these regions.



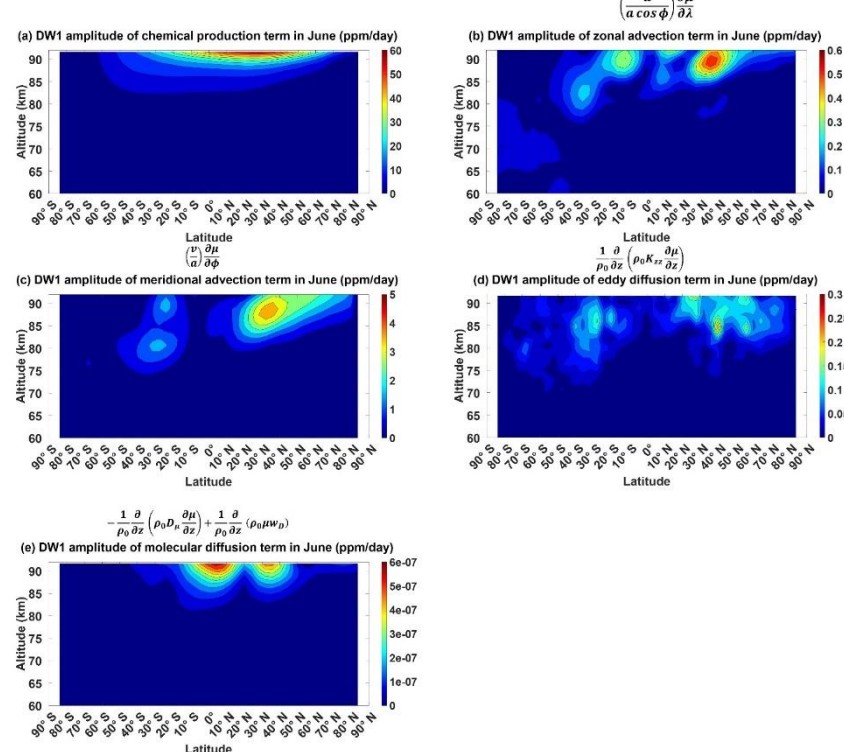

**Figure B2: Migrating Diurnal Tide component in June solstice of (a) CO's chemical production term, (b) CO's zonal advection term, (c) CO's meridional advection term, (d) CO's eddy diffusion term and (e) CO's molecular diffusion term. All are in units of ppm/day.**

## Appendix C: Interannual phenomena in $T'$

Figure C1 identifies interannual phenomena found in $h'_T$ and here we show that the features are very similar to that for $h'_\mu$ in figure 9. Figure C1A shows the time-series of MLS $h'_T$ at ~90 km, SD-WACCM-X $h'_T$ at ~92 km and the QBO index. Figure C1B shows the time-series of MLS $h'_T$ at ~90 km, SD-WACCM-X $h'_T$ at ~92 km and the MEI index. Figure C1C shows the cross-wavelet spectrum between MLS $h'_T$ and the QBO index. Figure C1C reveals that both MLS $h'_T$ and the QBO index have a statistically significant oscillation with periods of around 24 months. It is statistically significant between 2005 and 2018. The arrows are pointed slightly upward which indicate that MLS $h'_T$ peak slightly ahead of the QBO index. The arrows also indicate that MLS $h'_T$ increases during the westerly phase of the QBO while it decreases during the easterly phase of the QBO.





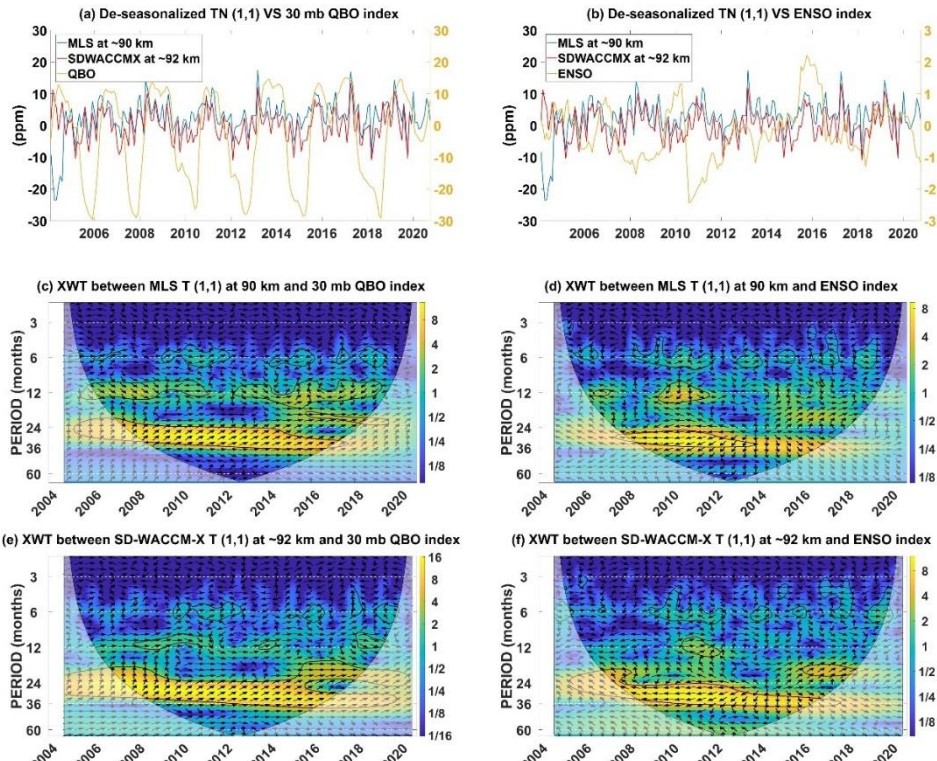

**Figure C1: (a) De-seasonalized time-series of MLS temperature (1,1) at ~90 km, SD-WACCM-X temperature (1,1) at ~92 km and the QBO index. (b) De-seasonalized time-series of MLS temperature (1,1) at ~90 km, SD-WACCM-X temperature (1,1) at ~92 km and the ENSO index. (c) Cross-wavelet spectrum between MLS temperature (1,1) at ~90 km and the QBO index. (d) Cross-wavelet spectrum between MLS temperature (1,1) at ~90 km and the ENSO index. (e) Cross-wavelet spectrum between SD-WACCM-X temperature (1,1) at ~92 km and the QBO index. (f) Cross-wavelet spectrum between SD-WACCM-X temperature (1,1) at ~92 km and the ENSO index.**

Figure C1D shows the cross-wavelet spectrum between MLS $h'_T$ and the MEI index. This spectrum reveals that both MLS $h'_T$ and the MEI index have a statistically significant oscillation of around 30 months between 2008 and 2012. This does coincide with a strong La Nina event. The arrows are pointed almost fully to the left which indicate that MLS $h'_T$ and ENSO are anti-correlated. MLS $h'_T$ increases during La Nina and it decreases during El Nino.

Figure C1E shows the cross-wavelet spectrum between SD-WACCM-X $h'_T$ and the QBO index. This spectrum reveals that both SD-WACCM-X $h'_T$ and the QBO index have a statistically significant oscillation with periods ranging from 20 to 30 months and the statistical significance is found throughout all years. From 2015 to 2020, both have a statistically significant oscillation with period around 15 to 20 months. Like MLS $h'_T$, the arrows are pointed slightly upward which indicate that SD-WACCM-X $h'_T$ peak slightly ahead of the QBO index and that SD-WACCM-X $h'_T$ increases during the westerly phase of the QBO while it decreases during the easterly phase of the QBO.

Figure C1F shows the cross-wavelet spectrum between SD-WACCM-X $h'_T$ and the MEI index. This spectrum reveals that both have a statistically significant oscillation with period of around 24 to 36 months. This period of statistical significance



lasts from 2006 to 2016. Like MLS $h'_T$, the arrows are pointed almost fully to the left which indicate that SD-WACCM-X $h'_T$ and ENSO are almost anti-correlated and that SD-WACCM-X $h'_T$ increases during La Nina and it decreases during El Nino.

**Code Availability**

As a component of the Community Earth System Model, WACCM-X source code are publicly available at http://www.cesm.ucar.edu.

**Data Availability**

The SABER dataset presented in this paper are accessible from the SABER website: http://saber.gats-inc.com/data.php. The MLS dataset presented in this paper are accessible from the MLS website: https://aura.gsfc.nasa.gov/mls.html. QBO index is accessible from http://www.cpc.ncep.noaa.gov/data/indices/qbo.u30.index. ENSO/MEI index is accessible from https://psl.noaa.gov/enso/mei/.

**Author contributions**

Conceptualization and investigation were done by CCJS, DLW and JNL. Formal Analysis and visualization were done by CCJS with help and supervision from DLW and JNL. Data curation on MLS data was done by JNL. Data curation on SABER data was done by CCJS. Access to SD-WACCM-X was provided by LQ and HL. SD-WACCM-X was run on Cheyenne (doi:10.5065/D6RX99HX) provided by NCAR's Computational and Information Systems Laboratory, sponsored by the 755 National Science 510 Foundation. DLW, JNL and LCC provided funding acquisition. CCJS wrote the original draft of the manuscript with help from DLW and JNL. All authors reviewed and edited the manuscript.

**Competing Interests**

The authors declare that they have no conflict of interest.

**Acknowledgements**

CCJS and LCC acknowledge Taiwan National Science and Technology Council grants 111-2636-M-008-004, 107-2923-M-008-001-MY3, and 110-2923-M-008-005-MY3, as well as the Higher Education SPROUT grant to the Center for Astronautical Physics and Engineering from the Taiwan Ministry of Education. The work of DLW and JNL was supported by NASA's TSIS Project and Sun-Climate research. LQ acknowledges support from the following NASA grants: 80NSSC19K0278, 80NSSC20K0189, NNH19ZDA001N-HGIO and NNH19ZDA001N-HSR. HL acknowledges support from 765 NASA grant 80NSSC20K1323. National Center for Atmospheric Research is a major facility sponsored by the National Science Foundation under Cooperative Agreement No. 1852977.





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
