# Peer review of "Aura/MLS observes, and SD-WACCM-X simulates the seasonality, quasi-biennial oscillation and El Nino Southern Oscillation of the migrating diurnal tide driving upper mesospheric CO primarily through vertical advection"

_Atmospheric Chemistry and Physics, 2022_

## Referee Comment (RC1)

Review of "Aura/MLS observes, and SD-WACCM-X simulates the seasonality, quasi-biennial oscillation and El Nino Southern Oscillation of the migrating diurnal tide driving upper mesospheric CO primarily through vertical advection" by Salinas et al.

Summary:

This paper presents results from two observational datasets and a numerical simulation to show that diurnal variations in CO observed by MLS are the result of vertical advection by the migrating diurnal tide. The paper then goes on to show consistency of this interpretation with tidal variations on 6-month, QBO, and ENSO time scales.

The results are interesting. The authors should be commended for recognition that key information about atmospheric tides can be found in observations of trace species in the atmosphere and for then extracting the signal. The signal variation with season and with the QBO is also a useful contribution. The demonstration that observed diurnal variations in a chemical field are due to transport rather than photochemistry is also a strength of this investigation.

However, the paper needs some work. The interpretation of results, especially the confusion about how the perturbations in temperature and CO are related to DW1 (major comment #1, below), leads to misleading claims about the tidal impacts. My primary concerns are laid out in the comments below, along with suggestions to address them. Of the major comments, please take particular care to address #1 and #8.

Recommendation: major revisions needed

Major comments:

1. There will be some readers who are not familiar with the analysis method of subtracting observations from the ascending and descending parts of the orbit to get the diurnal variation. It is important that you explain to them how to interpret these results. In particular, a maximum or minimum of this value does **not** indicate a maximum or minimum in the tidal amplitude. The phase of the actual 3-d tide varies with altitude. At the point where the local times of the observations differ from those of the daily minimum and maximum of the tide for that particular altitude by $+\pi/2$ or $-\pi/2$ (i.e. by 6 hours), the ascending minus descending difference will be identically zero, regardless of the tidal amplitude. A difference in the altitude at which the T' or $\mu'$ goes to zero in different datasets means that the tide phases are different but, by itself, does not give any information about the relative amplitudes of DW1 in the datasets being compared. This confusion begins in the abstract and is seen throughout the paper. For example, line 85 labels the feature as "DW1-induced perturbations", which is not inaccurate although not completely clear, but later this distinction is dropped. Line 144 uses the misleading term "DW1 component of CO and temperature". To ensure that readers are not confused, I recommend not to use the term DW1 at all. Referring to the signals as T' and $\mu'$ is okay as long as they are well-defined and carefully explained.

2. It is not clear why the SABER results are included. As noted in the paper, SABER can resolve DW1 in temperature. However, the actual tidal structure is never compared with the field

referred to as DW1 in the paper, which is not the actual tide (see comment above). Such a comparison could be useful to illustrate the relation between the tide itself and the perturbations analyzed in the paper.

3. The term chemical lifetime is used in a manner inconsistent with common usage. The lifetime normally is the time it takes for a molecule to decay due to photochemistry, not the timescale for it to be generated. Relevant to the discussion (lines 30-32; Section 5.1), the production of CO actually has a diurnal timescale (produced only during daylight) but its lifetime is defined by its relatively slow chemical loss. It appears that Eq. (2) does not even include a loss term.

4. At several points (e.g., lines 122-126; 204-205; 442-443), you speculate that discrepancies between MLS and SABER observations or WACCM are due to aliasing by the semidiurnal tide. While this is likely a contributing factor, there are other contributors that could also be important. One is the much poorer vertical resolution of MLS in the upper mesosphere. Since DW1 has a vertical wavelength of ~25-30 km, the MLS field of view, which smears the observations over ~9-12 km, can substantially reduce the amplitude.

5. Another contributor to discrepancies between MLS and other data is a result of the change in local time of the MLS measurements with latitude. As the Aura orbit approaches the turn-around latitudes near the poles, the times of ascending and descending observations get closer together and eventually coincide. Therefore the differences between them are less and less representative of the diurnal tide. Around 70-75°, the time differences are closer to 6 than to 12 hours. The average of ascending and descending times in high northern latitudes is opposite (12 hours different) to that in high southern latitudes. It is therefore no wonder that the MLS latitude structure at high latitudes in Figure 3 does not look like that of the (1,1) Hough mode. Attention to these local times could be relevant to the discussion of hemispheric differences (lines 596-601; 609-610).
In addition to revising the discussion to take the confusing latitude variation into account, I suggest to redo the fit of the results to the (1,1) Hough function but using a limited latitude range such as only latitudes equatorward of +/-40° or +/-50°.

6. Related to the previous comment, the perturbation in μ' due to tidal transport depends not just on the tidal winds but also on the vertical gradient of the mean composition. Don't lose sight of this when you are interpreting differences; see in particular the discussion of the hemispheric distortions seen in Figures 3 and 5.

7. (Figure 10c-10f and discussion) You are not alone in splitting time series into two or more segments to improve the apparent correlation. However, this should not be done unless you can provide a valid reason, based on physics, for doing so. Even the full 18-year timeseries is very short for identifying variations with the timescale of ENSO. What I see from your results is that there is no consistent impact of ENSO on your tidally influenced variables. The obvious interpretation is that the coincidences that appear for a few years are due to random interannual variability, not a causal mechanism. Leave it that way rather than trying to force small segments of the data into an agreement. I recommend that you drop the ENSO comparisons and discussion completely and merely mention that no consistent relation was found.

Minor comments

1. (lines 58-60) There are other observational studies that have shown a QBO in diurnal tidal winds. See Burrage et al. (1995), Pramitha et al. (2021), Xu et al. (2009).
2. Section 2 indicates that the data are on pressure levels but all results are presented as a function of altitude. How was the altitude determined? Did you actually interpret the observations and model output to altitude or do you use an approximate altitude such as the global & seasonal average or a fixed log-pressure to altitude ratio? Were the Hough mode fits done on altitude or pressure levels?
3. Section 5.1 is confusing. What function are the terms being fit to with the least squares fit?
4. (lines 309-310) "… if the vertical gradient of a tracer's daily-mean zonal-mean component is positive (the gradient increases with height) …": omit the words in parentheses if you mean that the tracer increases with height, not its gradient.
5. Check the sentence beginning at line 320; this looks like it's backwards.
6. I could not follow any part of the paragraph beginning at line 407. Why do you say that MLS CO $h_\mu'$ and SABER $h_T'$ are weaker? The magnitudes shown in the plots look larger. The second sentence is somewhat garbled so it is not clear what point is intended by these comparisons.
7. (line 426ff) "Above 90 km, their seasonality shifts into having a primary peak close to June solstice." This shift could be because the phase of DW1 is such that the 2 AM and 2 PM differences due to the tide itself are small (as in my first major comment). For example, a seasonal shift in the tide phase could contribute. A look at the phase of the full DW1 from SABER could help with the Interpretation.
8. (Figure 9) The phase lags can be reduced or eliminated by using a different pressure level for your QBO index. This might improve the MLR analyses shown later.
9. Please include references for the data you use for the QBO and ENSO indices.

Reference not cited in the manuscript

Burrage, M. D., Hagan, M. E., Skinner, W. R., Wu, D. L., and Hays, P. B., 1995. Long-term variability in the solar diurnal tide observed by HRDI and simulated by the GSWM, Geophys. Res. Lett., 22,2641–2644, doi:10.1029/95GL02635, 1995.

---

## Referee Comment (RC2)

Review of *"Aura/MLS observes, and SD-WACCM-X simulates the seasonality, quasi-biennial oscillation and El Nino Southern Oscillation of the migrating diurnal tide driving upper mesospheric CO primarily through vertical advection"*, by Salinas et al.

Recommendation: Revisions.

This paper reports the morphology and long-term variations in the diurnal cycles of T and CO observed by MLS, and extracted from an SD run of WACCM-X. The objective is to determine whether CO can be interpreted as a passive tracer of tidal motion. The authors demonstrate that the structures of diurnal CO and T closely track each other in both the data and in WACCM-X. By computing the mass budget of CO in WACCM-X, they are able to attribute the presence of tidal CO to vertical advection. The diurnal CO is also found to vary at QBO and ENSO periods.

This paper presents new information in the form of diurnal CO analyses, performs useful cross validation among MLS, SABER and WACCM-X T and CO, establishes the role of vertical tracer advection for tides, and reinforces earlier findings of QBO and ENSO variability in the propagating diurnal tide. Publication is therefore recommended following the revisions suggested below.

1. Lines 78-83: Does MLS sample at 2AM and 2PM at all latitudes? A latitude versus local time map might be helpful.
2. Sections 3 and 4: Figures 2 and 3 are described in exhaustive, almost mind-numbing detail. Instead of listing the altitude and latitude of every positive and negative extremum in each panel, I suggest a more concise wording with the goal of leaving the reader with the following "take-home" messages:

   a. The structures are dominated by (1,1) in March. A line plot of the (1,1) mode would be useful here.
   b. WACCM-X DW1 exhibits an additional "pulse" above 90 km in March that is not seen in MLS, both in CO and T, due to either a shorter vertical wavelength in WACCM-X, or to a phase offset between the model and the data.
   c. Patterns of T and CO are more asymmetric in June than in March. Please lose the "distortion of (1,1)" terminology. (See comment 8 below.)

3. Figures 2 and 3 have a lot of relatively empty space in them, with the interesting features crowded above 85 km. I suggest replotting them with the vertical axis starting at 75 km.
4. The chaotic middle and high latitude features in T and CO during winter months probably reflect variations in the zonal mean T and CO, instead of tides.
5. Line 194: Rewrite as "Although the latitude structure of DW! MLS CO $\mu'$ and SD-WACCM-X CO $\mu'$ have similarities to the DW1 temperature…".
6. Line 196: Rewrite as "…later use this to prove that the DW1 affects CO."
7. Lines 204 and 224: "aliasing of other tidal components into MLT T' and CO". I suggest being more specific here. Mention aliasing of migrating semidiurnal tides if the asc-desc LT difference is not 12 hours; also, are you thinking of terdiurnal tide leakage?
8. Lines 228-229, 240, 249, 607: These areas of the paper all refer to "distorted" of the (1,1) mode.

   (1,1) is an immutable eigenmode, characterized by a maximum at the equator, minima around 24N and 24S, and a uniform vertical wavelength of ~27 km. If the global structure of the tide deviates from (1,1) this is not due to "distortion" of (1,1), but the presence of additional Hough modes such as (1,2), (1,-1), etc.

9. Lines 230-231: The Forbes, McLandress, and Mukhartov papers cited do not discuss any relationship between the tides and the wave-driven residual mean circulation (v*,w*). Do you mean to say "zonally averaged winds"?
10. Lines 239: Delete the reference to nonmigrating tides in the aliasing discussion, as they do not alias to the zonal mean or the migrating tides. Nonmigrating tides do not alias into the zonal mean.
11. Provide a reference for equation 2.

    How is the DW1 component of the nonlinear terms defined? Do they arise from the advection of the DW1 components of $\mu$ by zonally averaged (u,v,w)? Or is it advection of time-mean $\mu$ by the tidal (u,v,w)?

12. Equation 3: This equation and its physical basis needs to be explained. I did not see any obvious analogies with the expressions in Eckermann et al. 1998. Since vertical motion does not appear, I presume it is inferred

adiabatically from T' through $\partial T/\partial t = N^2 w'$. Is this correct? For tidal motions, why does the frequency not appear in equation 3?

13. Lines 307-322. This section is much too wordy and repetitive. Since the vertical gradient of time mean $\mu$ is positive in the upper mesosphere (as seen in Figure 1), we don't need to read through hypothetical negative time-mean gradient scenarios. This entire segment can be summarized as:

"Equation 3 indicates that when the vertical gradient of the time-mean zonal mean $\mu$ is positive, then an increase in $\mu'$ requires T' > 0, which under adiabatic conditions implies a net downwelling. Conversely, a decrease in $\mu'$ implies T < 0', and net adiabatic upwelling."

Line 327 and 330: Replace "good" with "positive".

Lines 331-333: "For both MLS CO $\mu'$ and SD-WACCM-X CO $\mu'$, figures 4c and 4d indicate that the positive perturbations are driven by a relative downwelling due to the DW1 tide while the negative perturbations are driven by a relative upwelling."

Since we are not shown either w or $\partial u/\partial z$, there is no way to deduce vertical motion information from anything in Figure 4. Either show these variables, or remove this sentence.

14. Lines 372, 416, 417, and page: Replace "regress" with "project". "We project the latitude profiles of CO $\mu'$ onto the (1,1) Hough mode profile.

15. Line 407: "Figures 6a and 6b showed MLS CO h' is weaker than SD-WACCM-X CO h'.

Actually, MLS looks stronger than WACCM-X to me.

16. Figures 7a-c and 8a-c are difficult to read in general, and certainly for more nuanced features such as "Above 90 km, their seasonality shifts into having a primary peak close to June solstice". I recommend staring the vertical axis at 75 or 80 km, or presenting the main features as line plots at selected representative altitudes.

17. Lines 480, 511, 513: CO h' increases…"

What are the units of Figures 9c-f? Amplitude? Correlation?

What aspects of h' and $h_\mu$ "increase"

18. Line 493: Change "of temperature" to "tide".
19. Line 514: "Most studies have found that the (1,1) mode should decrease during El Nino events".

    In fact, Lieberman et al. (2007) showed that (1,1) *increased* during ENSO events. The reason is that the climatological dry tongue disappears during the El Nino phase, leading to a more longitudinally uniform water vapor distribution, and therefore a stronger (1,1) forcing by water vapor heating.

20. Section 7: The Summary is much too long, and repeats details that were already worked over in the main body of the paper. The entire section can be condensed to:

    "This work uses 17 years of CO observations provided by the Microwave Limb Sounder (MLS) on-board the Aura satellite to analyse the seasonal and interannual variability of the DW1 component of upper mesospheric CO. These were then compared to simulations by the Specified Dynamics – Whole Atmosphere Community Climate Model with Ionosphere/Thermosphere eXtension (SD-WACCM-X). CO DW1 is dominated by the (1,1) mode in both MLS data and WACCM-X. However, MLS only observes one pulse of the (1,1) mode between 80 km and 95 km while SD-WACCM-X simulates two pulses. This could be due to MLS' limited vertical resolution, or it could be due to inaccuracies in SD-WACCM-X simulation of the background atmosphere and/or tidal vertical propagation.

    The model-data comparison revealed that the structure of upper mesospheric MLS CO's DW1 component is primarily driven by DW1-induced vertical advection over all latitudes during equinox seasons, and over all latitudes except the winter middle to high latitudes during solstice seasons. This could suggest that MLS CO's DW1 component over the winter middle to high latitudes may be driven by other mechanisms such as meridional advection, eddy diffusion and/or chemistry. It could also suggest that the data over the winter middle to high latitudes may be affected by inadequate sampling.

In addition, we find that the interannual variability of MLS CO (1,1) and SD-WACCM-X CO (1,1) is primarily driven by the QBO and ENSO's effects on DW1- induced vertical advection. These conclusions suggest that we can use CO as a tracer for vertical advection due to the DW1 tide and the (1,1) mode on seasonal and interannual timescales. **"**

**Grammar and style:**

1. Line 40: New paragraph at "While".
2. Line 97: New paragraph at "Model".
3. Pages 11-12 are a bit too verbose. Consider deleting line 302 (If CO $\mu'$ and CO $\mu'$ are similar, then we can argue that vertical advection does primarily drive CO $\mu'$) and lines 308-312 (Equation 2 indicates…)
4. Line 370-371: Rewrite as "In this section, we examine seasonal and interannual variations in the (1,1) mode of CO."
5. Line 378: New paragraph at "Figure 6".
6. Line 446-459: "For example, Smith et al (2010) proved… very similar but for mesospheric SABER water vapor." Delete, unnecessary verbiage.
7. Line 477: New paragraph at "Figure 9".
8. Line 565: New paragraph at "Figure 10b".

---

## Author Comment (AC1)

We wish to thank the reviewer for their helpful comments. We have modified the manuscript as suggested. Below shows our responses to all the comments. Reviewer's comments are in bold red while our responses are in black. Note that, unless otherwise specified, all line numbers mentioned in the responses to comments refer to the numbers in the new (no tracking) manuscript.

**REVIEWER 1:**

**Review of "Aura/MLS observes, and SD-WACCM-X simulates the seasonality, quasi-biennial oscillation and El Nino Southern Oscillation of the migrating diurnal tide driving upper mesospheric CO primarily through vertical advection" by Salinas et al.**

**Summary:**

**This paper presents results from two observational datasets and a numerical simulation to show that diurnal variations in CO observed by MLS are the result of vertical advection by the migrating diurnal tide. The paper then goes on to show consistency of this interpretation with tidal variations on 6-month, QBO, and ENSO time scales. The results are interesting. The authors should be commended for recognition that key information about atmospheric tides can be found in observations of trace species in the atmosphere and for then extracting the signal. The signal variation with season and with the QBO is also a useful contribution. The demonstration that observed diurnal variations in a chemical field are due to transport rather than photochemistry is also a strength of this investigation. However, the paper needs some work. The interpretation of results, especially the confusion about how the perturbations in temperature and CO are related to DW1 (major comment #1, below), leads to misleading claims about the tidal impacts. My primary concerns are laid out in the comments below, along with suggestions to address them. Of the major comments, please take particular care to address #1 and #8.**

**Recommendation: major revisions needed**

**Major comments:**

1. **There will be some readers who are not familiar with the analysis method of subtracting observations from the ascending and descending parts of the orbit to get the diurnal variation. It is important that you explain to them how to interpret these results. In particular, a maximum or minimum of this value does *not* indicate a maximum or minimum in the tidal amplitude. The phase of the actual 3-d tide varies with altitude. At the point where the local times of the observations differ from those of the daily minimum and maximum of the tide for that particular altitude by +p/2 or -p/2 (i.e. by 6 hours), the ascending minus descending difference will be identically zero, regardless of the tidal amplitude. A difference in the altitude at which the T' or μ' goes to zero in different datasets means that the tide phases are different but, by itself, does not give any information about the relative amplitudes of DW1 in the datasets being compared. This confusion begins in the abstract and is seen throughout the paper. For example, line 85 labels the feature as "DW1-induced perturbations", which is not inaccurate although not completely clear, but later this distinction is dropped. Line 144 uses the misleading term "DW1 component of CO and temperature". To ensure that**

**readers are not confused, I recommend not to use the term DW1 at all. Referring to the signals as T' and μ' is okay as long as they are well-defined and carefully explained.**

We first clarify that we do not simply subtract the ascending and descending parts of the orbit. It is not enough that a data point is part of the ascending and descending parts of the orbit. We make sure to check that the values are at around ~2 AM and ~2 PM local-times when calculating these zonal-means.

In figure 2, we have added the following sentences in lines 161 – 165 explaining how to interpret the results: *"When a dataset has full local-time coverage, figure 2 would come in the form of amplitude contour maps and it would be accompanied by a phase contour map. The amplitude map will then clearly indicate where exactly the tides are strongest. In contrast, figure 2 and the other figures showing $\mu'$ cannot indicate where exactly the tidal amplitudes are strongest. It can only indicate where the tide significantly affects CO (tidal perturbations) but it cannot indicate the relative strength of this influence."*

We agree with the concerns on the use of the term "DW1". After re-checking the manuscript, we first point out that the first parts of the results section do not mention the term DW1. We carefully only used the terms $T'$ and $\mu'$. This is because we acknowledge that in these sections, we have not yet established that DW1 predominantly drives $T'$ and $\mu'$. In the next section though, we have established through a comparison between MLS and SABER that $T'$ may be predominantly driven by DW1. We complemented this by mentioning numerous previous papers establishing this. Then, the next section shows that the DW1 driving $T'$ may also be driving $\mu'$. Despite that, the reminder of the results section only use the terms $T'$, $\mu'$ or (1,1). We do not mention anything about the DW1 tide although the term pops up in the section-headers. It is only in the summary and conclusions section that after all the results, we can conclude that much of $T'$ and $\mu'$ is primarily driven by DW1. As for the abstract, we changed "DW1-induced perturbations" to "local-time perturbations".

2. **It is not clear why the SABER results are included. As noted in the paper, SABER can resolve DW1 in temperature. However, the actual tidal structure is never compared with the field referred to as DW1 in the paper, which is not the actual tide (see comment above). Such a comparison could be useful to illustrate the relation between the tide itself and the perturbations analyzed in the paper.**

Section 4 compares SABER $T'$ with MLS $T'$. Note that SABER $T'$ is the DW1 component of SABER $T$ because the DW1 amplitudes and phases were first calculated before calculating SABER $T'$. In this section, we do show there is good agreement between SABER $T'$ and MLS $T'$ giving additional confidence that MLS $T'$ may be predominantly driven by DW1.

3. **The term chemical lifetime is used in a manner inconsistent with common usage. The lifetime normally is the time it takes for a molecule to decay due to photochemistry, not the timescale for it to be generated. Relevant to the discussion (lines 30-32; Section 5.1), the production of CO actually has a diurnal timescale (produced only during daylight) but its lifetime is defined by its relatively slow chemical loss. It appears that Eq. (2) does not even include a loss term.**

We have modified lines 30-32 to define the term chemical timescales and avoid using the term chemical lifetime. The new lines are: *"This reaction makes the timescales of the chemical*

*reactions driving CO's variability (hereafter referred to as chemical timescales) longer than dynamical timescales (Minschwaer et al, 2010)."* We have also included the loss term in equation 2 as well as in the Appendix B discussions.

4. **At several points (e.g., lines 122-126; 204-205; 442-443), you speculate that discrepancies between MLS and SABER observations or WACCM are due to aliasing by the semidiurnal tide. While this is likely a contributing factor, there are other contributors that could also be important. One is the much poorer vertical resolution of MLS in the upper mesosphere. Since DW1 has a vertical wavelength of ~25-30 km, the MLS field of view, which smears the observations over ~9-12 km, can substantially reduce the amplitude.**

We don't change lines 122-126 (original manuscript) because we use that to lead to our rationale behind the use of SABER observations in the next paragraph.

But for lines 204-205 (original manuscript), we have added the following sentence: *"It may also be attributed to differences in the instruments' vertical resolutions. SABER has a vertical resolution of ~2 km while MLS has a vertical resolution of ~10 km (Remsberg et al., 2008; Livesey et al., 2011). Given that DW1 typically has a vertical wavelength of ~25-30 km, MLS' coarser vertical resolution can substantially reduce the amplitudes."*

For lines 442-443 (original manuscript), we have modified the lines into: *"Figures 6a, 6c and 6d shows that the seasonality of MLS CO $h'_\mu$ may be affected by the incomplete local-time sampling of MLS or its coarse vertical resolution."*

5. **Another contributor to discrepancies between MLS and other data is a result of the change in local time of the MLS measurements with latitude. As the Aura orbit approaches the turnaround latitudes near the poles, the times of ascending and descending observations get closer together and eventually coincide. Therefore the differences between them are less and less representative of the diurnal tide. Around 70-75°, the time differences are closer to 6 than to 12 hours. The average of ascending and descending times in high northern latitudes is opposite (12 hours different) to that in high southern latitudes. It is therefore no wonder that the MLS latitude structure at high latitudes in Figure 3 does not look like that of the (1,1) Hough mode. Attention to these local times could be relevant to the discussion of hemispheric differences (lines 596-601; 609-610). In addition to revising the discussion to take the confusing latitude variation into account, I suggest to redo the fit of the results to the (1,1) Hough function but using a limited latitude range such as only latitudes equatorward of +/-40° or +/-50°.**

As mentioned in our reply to comment #1, we first clarify that we do not subtract the ascending and descending parts of the orbit. We are subtracting values at ~2AM and values at ~2PM. Hence, a contributor to the discrepancies that we point out relates to the uneven sampling across latitudes. While we are confident in the adequate sampling over the low-latitudes, this may not be the case over the mid-latitudes. This is alluded to, for example, line 235-240: *"These differences between MLS $T'$ and SABER $T'$ over the mid-latitudes may be a result of MLS inadequate sampling causing significant aliasing from other tides."*

However, we did redo the fits to only use values equatorward of +/- 50 degrees. The values didn't change significantly but these new values are the ones used in the revised manuscript's plots.

6. **Related to the previous comment, the perturbation in μ' due to tidal transport depends not just on the tidal winds but also on the vertical gradient of the mean composition. Don't lose sight of this when you are interpreting differences; see in particular the discussion of the hemispheric distortions seen in Figures 3 and 5.**

This is immediately mentioned in the paragraphs following equation 3 and, in the results section, we use this to argue whether the perturbations are due to a net downwelling or net upwelling (lines 335-336 and lines 364-365).

7. **(Figure 10c-10f and discussion) You are not alone in splitting time series into two or more segments to improve the apparent correlation. However, this should not be done unless you can provide a valid reason, based on physics, for doing so. Even the full 18-year timeseries is very short for identifying variations with the timescale of ENSO. What I see from your results is that there is no consistent impact of ENSO on your tidally influenced variables. The obvious interpretation is that the coincidences that appear for a few years are due to random interannual variability, not a causal mechanism. Leave it that way rather than trying to force small segments of the data into an agreement. I recommend that you drop the ENSO comparisons and discussion completely and merely mention that no consistent relation was found.**

We have removed figures 10c to 10f as well as the pertinent discussions. We have also modified the first few sentences of section 6.4 into: "The previous sub-section found that QBO and ENSO variabilities are present in both MLS CO $h_\mu'$ and SD-WACCM-X CO $h_\mu'$. In this section, we quantify the changes in MLS CO $h_\mu'$ or SD-WACCM-X CO $h_\mu'$ due to QBO. We don't quantify the changes due to ENSO because there were only a few events during our data-span. Hence, any estimated response may be biased."

**Minor comments**

**1. (lines 58-60) There are other observational studies that have shown a QBO in diurnal tidal winds. See Burrage et al. (1995), Pramitha et al. (2021), Xu et al. (2009).**

Xu et al (2009) and Pramitha et al (2021) are already there. We added Burrage et al. (1995).

**2. Section 2 indicates that the data are on pressure levels but all results are presented as a function of altitude. How was the altitude determined? Did you actually interpret the observations and model output to altitude or do you use an approximate altitude such as the global & seasonal average or a fixed log-pressure to altitude ratio? Were the Hough mode fits done on altitude or pressure levels?**

All calculations are done on pressure levels but for the plots, we replaced the pressure levels with their approximate altitudes.

**3. Section 5.1 is confusing. What function are the terms being fit to with the least squares fit?**

After presenting equation 2, we clarified the fit by adding the sentence: *"The DW1 component of each term is calculated by fitting the terms into the equation* $X(\lambda, t) = \bar{X} + \hat{X}_{n,s} \cos(\pi t/24 - (-1)\lambda - \hat{\psi}_{n,s})$ *using 2D least-squares fit."*

**4. (lines 309-310) "… if the vertical gradient of a tracer's daily-mean zonal-mean component is positive (the gradient increases with height) …": omit the words in parentheses if you mean that the tracer increases with height, not its gradient.**

Omitted.

**5. Check the sentence beginning at line 320; this looks like it's backwards.**

Corrected.

**6. I could not follow any part of the paragraph beginning at line 407. Why do you say that MLS CO hμ' and SABER hT' are weaker? The magnitudes shown in the plots look larger. The second sentence is somewhat garbled so it is not clear what point is intended by these comparisons.**

We have changed this entire paragraph into: "Figures 6a and 6b showed MLS CO $h'_\mu$ is stronger than SD-WACCM-X CO $h'_\mu$. Figures 6d and 6e also showed that SABER $h'_T$ is stronger than SD-WACCM-X $h'_T$. A larger MLS CO $h'_\mu$ than simulated is consistent with a larger realistic MLS or SABER $h'_T$ than simulated. An underestimation of SD-WACCM-X $h'_T$ indicates inaccuracies in the simulated background atmosphere, tidal source or tidal dissipation mechanisms."

**7. (line 426ff) "Above 90 km, their seasonality shifts into having a primary peak close to June solstice." This shift could be because the phase of DW1 is such that the 2 AM and 2 PM differences due to the tide itself are small (as in my first major comment). For example, a seasonal shift in the tide phase could contribute. A look at the phase of the full DW1 from SABER could help with the Interpretation.**

We have added the following lines: *"This could suggest that the latitude structure of DW1's phase during solstice (equinox) causes maximum (minimum) values when taking the difference of values at ~2 AM and at ~2 PM. This consequently enhances (reduces) MLS CO* $h'_\mu$*. This is difficult to validate with a very high degree of uncertainty though even with SABER data because of the differences in MLS and SABER's vertical resolution."*

**8. (Figure 9) The phase lags can be reduced or eliminated by using a different pressure level for your QBO index. This might improve the MLR analyses shown later.**

Although the QBO affects other pressure levels, it primarily originates around ~30 mb. We do not use a QBO index in other pressure levels nor do we want to remove the phase lags because we want to point out that there is a noteworthy lag which indicates some form of wave-mean flow filtering could be involved.

**9. Please include references for the data you use for the QBO and ENSO indices.**

These have been specified in the "Data Availability" section.

**Reference not cited in the manuscript**

**Burrage, M. D., Hagan, M. E., Skinner, W. R., Wu, D. L., and Hays, P. B., 1995. Long-term variability in the solar diurnal tide observed by HRDI and simulated by the GSWM, Geophys. Res. Lett., 22,2641–2644, doi:10.1029/95GL02635, 1995.**

---

## Author Comment (AC2)

We wish to thank the reviewer for their helpful comments. We have modified the manuscript as suggested. Below shows our responses to all the comments. Reviewer's comments are in bold red while our responses are in black. Note that, unless otherwise specified, all line numbers mentioned in the responses to comments refer to the numbers in the new (no tracking) manuscript.

**REVIEWER 2:**

**Review of "Aura/MLS observes, and SD-WACCM-X simulates the seasonality, quasi-biennial oscillation and El Nino Southern Oscillation of the migrating diurnal tide driving upper mesospheric CO primarily through vertical advection", by Salinas et al.**

**Recommendation: Revisions.**

**This paper reports the morphology and long-term variations in the diurnal cycles of T and CO observed by MLS, and extracted from an SD run of WACCM-X. The objective is to determine whether CO can be interpreted as a passive tracer of tidal motion. The authors demonstrate that the structures of diurnal CO and T closely track each other in both the data and in WACCM-X. By computing the mass budget of CO in WACCM-X, they are able to attribute the presence of tidal CO to vertical advection. The diurnal CO is also found to vary at QBO and ENSO periods. This paper presents new information in the form of diurnal CO analyses, performs useful cross validation among MLS, SABER and WACCM-X T and CO, establishes the role of vertical tracer advection for tides, and reinforces earlier findings of QBO and ENSO variability in the propagating diurnal tide. Publication is therefore recommended following the revisions suggested below.**

**1. Lines 78-83: Does MLS sample at 2AM and 2PM at all latitudes? A latitude versus local time map might be helpful.**

To address this concern, we have added the following sentences: *"Nguyen and Palo (2013) have shown that up to around latitude 50 degrees, the data-points of MLS are at either ~2 AM or ~2 PM local-time. In our work, our calculations show that this can be extended up to latitudes ~80 degrees in both hemispheres although, the number of data-points aren't as much as over the low-latitudes. We make sure to note this in the analysis."*

**2. Sections 3 and 4: Figures 2 and 3 are described in exhaustive, almost mindnumbing detail. Instead of listing the altitude and latitude of every positive and negative extremum in each panel, I suggest a more concise wording with the goal of leaving the reader with the following "take-home" messages:**

> **a. The structures are dominated by (1,1) in March. A line plot of the (1,1) mode would be useful here.**

> **b. WACCM-X DW1 exhibits an additional "pulse" above 90 km in March that is not seen in MLS, both in CO and T, due to either a shorter vertical wavelength in WACCM-X, or to a phase offset between the model and the data.**

> **c. Patterns of T and CO are more asymmetric in June than in March. Please lose the "distortion of (1,1)" terminology. (See comment 8 below.)**

We have modified the presentation of figure 2 into:

[revised manuscript text omitted]

**3. Figures 2 and 3 have a lot of relatively empty space in them, with the interesting features crowded above 85 km. I suggest replotting them with the vertical axis starting at 75 km.**
Replotted with the vertical axis starting at 75 km.

**4. The chaotic middle and high latitude features in T and CO during winter months probably reflect variations in the zonal mean T and CO, instead of tides.**

In line 194, we added this sentence: *"This could suggest that MLS observes mean-flow changes affecting these structures that aren't simulated in the model."*

**5. Line 194: Rewrite as "Although the latitude structure of DW! MLS CO $\mu'$ and SD-WACCM-X CO $\mu'$ have similarities to the DW1 temperature…".**

Corrected.

**6. Line 196: Rewrite as "…later use this to prove that the DW1 affects CO."**

The entire sentence has been changed to: *"Although the latitude structure of DW1 MLS CO $\mu'$ and SD-WACCM-X CO $\mu'$ have similarities to the DW1 temperature, it has never been proven that the DW1 tide affects CO."*

**7. Lines 204 and 224: "aliasing of other tidal components into MLT T' and CO". I suggest being more specific here. Mention aliasing of migrating semidiurnal tides if the asc-desc LT difference is not 12 hours; also, are you thinking of terdiurnal tide leakage?**

We specify that the aliasing might be due to the migrating semidiurnal tides.

**8. Lines 228-229, 240, 249, 607: These areas of the paper all refer to "distorted" of the (1,1) mode. (1,1) is an immutable eigenmode, characterized by a maximum at the equator, minima around 24N and 24S, and a uniform vertical wavelength of ~27 km. If the global structure of the tide deviates from (1,1) this is not due to "distortion" of (1,1), but the presence of additional Hough modes such as (1,2), (1,-1), etc.**

This "distorted" term is first mentioned in the presentation of figure 2c. To clarify, we have added the following sentence: "By "distorted", we mean the presence of other diurnal Hough modes." We recognize that the common approach in other papers analyzing the (1,1) mode is to have placed quotes in the term "distorted".

**9. Lines 230-231: The Forbes, McLandress, and Mukhartov papers cited do not discuss any relationship between the tides and the wave-driven residual mean circulation (v*,w*). Do you mean to say "zonally averaged winds"?**

Yes. We have changed "winter residual circulation" into "zonally averaged winds".

**10. Lines 239: Delete the reference to nonmigrating tides in the aliasing discussion, as they do not alias to the zonal mean or the migrating tides. Nonmigrating tides do not alias into the zonal mean.**

Removed.

**11. Provide a reference for equation 2. How is the DW1 component of the nonlinear terms defined? Do they arise from the advection of the DW1 components of μ by zonally averaged (u,v,w)? Or is it advection of time-mean μ by the tidal (u,v,w)?**

We have cited Brasseur and Solomon (2006) for equation 2. In this analysis, we do not separate the linear and non-linear advection terms. We are just interested in determining the contributions

of total zonal advection, meridional advection, vertical advection, eddy diffusion, molecular diffusion, chemical production and chemical loss.

**12. Equation 3: This equation and its physical basis needs to be explained. I did not see any obvious analogies with the expressions in Eckermann et al. 1998. Since vertical motion does not appear, I presume it is inferred adiabatically from T' through ¶T/¶t = N2w'. Is this correct? For tidal motions, why does the frequency not appear in equation 3?**

We added the following brief derivation of the equation:

*"This equation is derived by first linearizing the continuity equation (equation 2). Then, we assume only the vertical advection term is important. Finally, we set all primed variables into the form $e^{i(kx-\sigma t)}$ where $k$ is the zonal wave number and $\sigma$ is the tidal frequency. This gives us this equation:*

$$i(k\bar{u} - \sigma)\mu'_w + \frac{\partial \bar{\mu}}{\partial z}w' = 0 \qquad\qquad (4)$$

*The same can be done to a form of the thermodynamic equation that assumes all temperature changes are due to adiabatic motion. This gives us this equation:*

$$i(k\bar{u} - \sigma)T' + Sw' = 0 \qquad\qquad (5)$$

*Combining equations 4 and 5 give equation 3."*

**13. Lines 307-322. This section is much too wordy and repetitive. Since the vertical gradient of time mean μ is positive in the upper mesosphere (as seen in Figure 1), we don't need to read through hypothetical negative time-mean gradient scenarios. This entire segment can be summarized as: "Equation 3 indicates that when the vertical gradient of the time-mean zonal mean μ is positive, then an increase in μ' requires T' > 0, which under adiabatic conditions implies a net downwelling. Conversely, a decrease in μ' implies T < 0', and net adiabatic upwelling."**

We've reduced these paragraphs into the following: *"Equation 3 indicates that if vertical advection does primarily drive a tracer's DW1 component and since figure 1 has shown that zonal-mean CO's vertical gradient is positive, CO μ' and T' are correlated. This also indicates that an increase in μ' requires T' > 0, which under adiabatic conditions implies a net downwelling. Conversely, a decrease in μ' implies T < 0', and net adiabatic upwelling."*

**Line 327 and 330: Replace "good" with "positive".**

Replaced.

**Lines 331-333: "For both MLS CO $\mu'$ and SD-WACCM-X CO $\mu'$, figures 4c and 4d indicate that the positive perturbations are driven by a relative downwelling due to the DW1 tide while the negative perturbations are driven by a relative upwelling." Since we are not shown either w or ¶u/¶z, there is no way to deduce vertical motion information from anything in Figure 4. Either show these variables, or remove this sentence.**

Yes, we are aware that we cannot deduce the exact or absolute vertical motion. Hence, we use term "relative".

**14. Lines 372, 416, 417, and page: Replace "regress" with "project". "We project the latitude profiles of CO μ′ onto the (1,1) Hough mode profile.**

Replaced.

**15. Line 407: "Figures 6a and 6b showed MLS CO h′ is weaker than SDWACCM-X CO h′. Actually, MLS looks stronger than WACCM-X to me.**

Corrected.

**16. Figures 7a-c and 8a-c are difficult to read in general, and certainly for more nuanced features such as "Above 90 km, their seasonality shifts into having a primary peak close to June solstice". I recommend staring the vertical axis at 75 or 80 km, or presenting the main features as line plots at selected representative altitudes.**

We have adjusted the vertical axis to begin at 75 km.

**17. Lines 480, 511, 513: CO h′ increases…" What are the units of Figures 9c-f? Amplitude? Correlation? What aspects of h' and hμ "increase"**

We clarify that the units of all cross-wavelet spectrum are in spectral power by adding the following line: *"In this and the succeeding spectra, encircled regions with the high spectral power correspond to oscillations statistically significant in both time-series (Grinsted et al, 2004)."*

To clarify what aspects of $h'_\mu$ increases or decreases, we add the following in lines 490: *"Depending on the arrows, one can deduce the correlations between CO $h'_\mu$ amplitude and QBO or ENSO. Consequently, the deduced correlation will imply whether CO $h'_\mu$ increases or decreases during, for example, westerly QBO phase."*

**18. Line 493: Change "of temperature" to "tide".**

Changed.

**19. Line 514: "Most studies have found that the (1,1) mode should decrease during El Nino events". In fact, Lieberman et al. (2007) showed that (1,1) increased during ENSO events. The reason is that the climatological dry tongue disappears during the El Nino phase, leading to a more longitudinally uniform water vapor distribution, and therefore a stronger (1,1) forcing by water vapor heating.**

We have modified this section to also include this suggested explanation: *"Most studies have found that the (1,1) mode should decrease during El Nino events. However, our results indicate that the effect of ENSO reversed during the 2015 El Nino. Kogure et al (2021) has explained this. Their work showed that the enhanced (1,1) tide in 2015 was a result of the overlapping occurrence of an easterly QBO phase and an El Nino event. Lieberman et al. (2007) also showed that the (1,1) mode increased during ENSO events because the climatological dry tongue disappears during the El Nino phase, leading to a more longitudinally uniform water vapor distribution, and therefore a stronger (1,1) forcing by water vapor heating. Our works adds to these previous studies by showing that MLS CO's (1,1) mode is also affected by ENSO in the same way."*

**20. Section 7: The Summary is much too long, and repeats details that were already worked over in the main body of the paper. The entire section can be condensed to: "
[revised manuscript text omitted]

**Grammar and style:**

**1. Line 40: New paragraph at "While".**

Corrected.

**2. Line 97: New paragraph at "Model".**

Corrected.

**3. Pages 11-12 are a bit too verbose. Consider deleting line 302 (If CO $\mu'$ and CO $\mu'$ are similar, then we can argue that vertical advection does primarily drive CO $\mu'$) and lines 308-312 (Equation 2 indicates…)**

As mentioned in major comment #13 above, we've reduced these paragraphs into the following: *"Equation 3 indicates that if vertical advection does primarily drive a tracer's DW1 component and since figure 1 has shown that zonal-mean CO's vertical gradient is positive, CO $\mu'$ and $T'$ are correlated. This also indicates that an increase in $\mu'$ requires $T' > 0$, which under adiabatic conditions implies a net downwelling. Conversely, a decrease in $\mu'$ implies $T < 0'$, and net adiabatic upwelling."*

**4. Line 370-371: Rewrite as "In this section, we examine seasonal and interannual variations in the (1,1) mode of CO."**

This suggested replacement oversimplifies what we intend to do in this section but we do reduce it into: *"In this section, we now focusing on determining vertical advection's impact on the seasonal and interannual variabilities of CO's (1,1) mode."*

**5. Line 378: New paragraph at "Figure 6".**

Corrected.

**6. Line 446-459: "For example, Smith et al (2010) proved… very similar but for mesospheric SABER water vapor." Delete, unnecessary verbiage.**

Removed.

**7. Line 477: New paragraph at "Figure 9".**

Corrected.

**8. Line 565: New paragraph at "Figure 10b".**

Corrected.